# CRYSTAL DIFFUSION VARIATIONAL AUTOENCODER FOR PERIODIC MATERIAL GENERATION

**Tian Xie,**[*] **Xiang Fu,**[*] **Octavian-Eugen Ganea,**[*] **Regina Barzilay, Tommi Jaakkola**
Computer Science and Artificial Intelligence Laboratory
Massachusetts Institute of Technology
Cambridge, MA 02139, USA
`{txie,xiangfu,oct,regina,tommi}@csail.mit.edu`

## ABSTRACT

Generating the periodic structure of stable materials is a long-standing challenge for the material design community. This task is difficult because stable materials only exist in a low-dimensional subspace of all possible periodic arrangements of atoms: 1) the coordinates must lie in the local energy minimum defined by quantum mechanics, and 2) global stability also requires the structure to follow the complex, yet specific bonding preferences between different atom types. Existing methods fail to incorporate these factors and often lack proper invariances. We propose a Crystal Diffusion Variational Autoencoder (CDVAE) that captures the physical inductive bias of material stability. By learning from the data distribution of stable materials, the decoder generates materials in a diffusion process that moves atomic coordinates towards a lower energy state and updates atom types to satisfy bonding preferences between neighbors. Our model also explicitly encodes interactions across periodic boundaries and respects permutation, translation, rotation, and periodic invariances. We significantly outperform past methods in three tasks: 1) reconstructing the input structure, 2) generating valid, diverse, and realistic materials, and 3) generating materials that optimize a specific property. We also provide several standard datasets and evaluation metrics for the broader machine learning community. [1]

## 1 INTRODUCTION

Solid state materials, represented by the periodic arrangement of atoms in the 3D space, are the foundation of many key technologies including solar cells, batteries, and catalysis (Butler et al., 2018). Despite the rapid progress of molecular generative models and their significant impact on drug discovery, the problem of material generation has many unique challenges. Compared with small molecules, materials have more complex periodic 3D structures and cannot be adequately represented by a simple graph like molecular graphs (Figure 1). In addition, materials can be made up of more than 100 elements in the periodic table, while molecules are generally only made up of a small subset of atoms such as carbon, oxygen, and hydrogen. Finally, the data for training ML models for material design is limited. There are only ∼200k experimentally known inorganic materials, collected by the ICSD (Belsky et al., 2002), in contrast to close to a billion molecules in ZINC (Irwin & Shoichet, 2005).

The key challenge of this task is in generating *stable* materials. Such materials only exist in a low-dimensional subspace of all possible periodic arrangements of atoms: 1) the atom coordinates must lie in the local energy minimum defined by quantum mechanics (QM); 2) global stability also requires the structure to follow the complex, yet specific

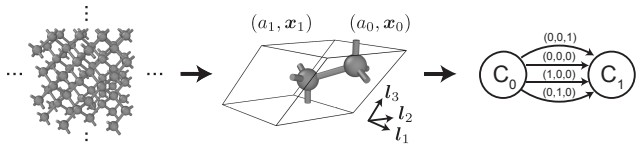

Figure 1: The periodic structure of diamond. The left shows the infinite periodic structure, the middle shows a unit cell representing the periodic structure, and the right shows a multi-graph (Xie & Grossman, 2018) representation.

---

[*]Equal contribution. Correspondence to: Tian Xie at txie@csail.mit.edu

[1]Code and data are available at `https://github.com/txie-93/cdvae`

bonding preferences between different atom types (section 3.2). The issue of stability is unique to material generation because valency checkers assessing molecular stability are not applicable to materials. Moreover, we also have to encode the interactions crossing periodic boundaries (Figure 1, middle), and satisfy permutation, translation, rotation, and periodic invariances (section 3.1). Our goal is to learn representations that can learn features of stable materials from data, while adhering to the above invariance properties.

We address these challenges by learning a variational autoencoder (VAE) (Kingma & Welling, 2014) to generate stable 3D materials directly from a latent representation without intermediates like graphs. The key insight is to exploit the fact that all materials in the data distribution are stable, therefore if noise is added to the ground truth structure, denoising it back to its original structure will likely increase stability. We capture this insight by designing a noise conditional score network (NCSN) (Song & Ermon, 2019) as our decoder: 1) the decoder outputs gradients that drive the atom coordinates to the energy local minimum; 2) it also updates atom types based on the neighbors to capture the specific local bonding preferences (e.g., Si-O is preferred over Si-Si and O-O in $SiO_2$). During generation, materials are generated using Langevin dynamics that gradually deforms an initial random structure to a stable structure. To capture the necessary invariances and encode the interactions crossing periodic boundaries, we use SE(3) equivariant graph neural networks adapted with periodicity (PGNNs) for both the encoder and decoder of our VAE.

Our theoretical analysis further reveals an intriguing connection between the gradient field learned by our decoder and an harmonic force field. De facto, the decoder utilizes the latter to estimate the forces on atoms when their coordinates deviate from the equilibrium positions. Consequently, this formulation provides an important physical inductive bias for generating stable materials.

In this work, we propose Crystal Diffusion Variational AutoEncoder (CDVAE) to generate stable materials by learning from the data distribution of known materials. Our main contributions include:

- We curate 3 standard datasets from QM simulations and create a set of physically meaningful tasks and metrics for the problem of material generation.
- We incorporate stability as an inductive bias by designing a noise conditional score network as the decoder of our VAE, which allows us to generate significantly more realistic materials.
- We encode permutation, translation, rotation, and periodic invariances, as well as interactions crossing periodic boundaries with SE(3) equivariant GNNs adapted with periodicity.
- Empirically, our model significantly outperforms past methods in tasks including reconstructing an input structure, generating valid, diverse, and realistic materials, and generating materials that optimize specific properties.

## 2 RELATED WORK

**Material graph representation learning.** Graph neural networks have made major impacts in material property prediction. They were first applied to the representation learning of periodic materials by Xie & Grossman (2018) and later enhanced by many studies including Schütt et al. (2018); Chen et al. (2019). The Open Catalyst Project (OCP) provides a platform for comparing different architectures by predicting energies and forces from the periodic structure of catalytic surfaces (Chanussot et al., 2021). Our encoder and decoder PGNNs directly use GNN architectures developed for the OCP (Klicpera et al., 2020b; 2021; Shuaibi et al., 2021; Godwin et al., 2021), which are also closely related to SE(3) equivariant networks (Thomas et al., 2018; Fuchs et al., 2020).

**Quantum mechanical search of stable materials.** Predicting the structure of unknown materials requires very expensive random search and QM simulations, and is considered a grand challenge in materials discovery (Oganov et al., 2019). State-of-the-art methods include random sampling (Pickard & Needs, 2011), evolutionary algorithms (Wang et al., 2012; Glass et al., 2006), substituting elements in known materials (Hautier et al., 2011), etc., but they generally have low success rates and require extensive computation even on relatively small problems.

**Material generative models.** Past material generative models mainly focus on two different approaches, and neither incorporate stability as an inductive bias. The first approach treats materials as 3D voxel images, but the process of decoding images back to atom types and coordinates often results in low validity, and the models are not rotationally invariant (Hoffmann et al., 2019; Noh et al., 2019; Court et al., 2020; Long et al., 2021). The second directly encodes atom coordinates,

types, and lattices as vectors (Ren et al., 2020; Kim et al., 2020; Zhao et al., 2021), but the models are generally not invariant to any Euclidean transformations. Another related method is to train a force field from QM forces and then apply the learned force field to generate stable materials by minimizing energy (Deringer et al., 2018; Chen & Ong, 2022). This method is conceptually similar to our decoder, but it requires additional force data which is expensive to obtain. Remotely related works include generating contact maps from chemical compositions (Hu et al., 2021; Yang et al., 2021) and building generative models only for chemical compositions (Sawada et al., 2019; Pathak et al., 2020; Dan et al., 2020).

**Molecular conformer generation and protein folding .** Our decoder that generates the 3D atomic structures via a diffusion process is closely related to the diffusion models used for molecular conformer generation (Shi et al., 2021; Xu et al., 2021b). The key difference is that our model does not rely on intermediate representations like molecular graphs. G-SchNet (Gebauer et al., 2019) is more closely related to our method because it directly generates 3D molecules atom-by-atom without relying on a graph. Another closely related work is E-NFs (Satorras et al., 2021) that use a flow model to generate 3D molecules. In addition, score-based and energy-based models have also been used for molecular graph generation (Liu et al., 2021) and protein folding (Wu et al., 2021). Flow models have also been used for molecular graph generation (Shi et al., 2020; Luo et al., 2021). However, these generative models do not incorporate periodicity , which makes them unsuitable for materials.

# 3 PRELIMINARIES

## 3.1 PERIODIC STRUCTURE OF MATERIALS

Any material structure can be represented as the periodic arrangement of atoms in the 3D space. As illustrated in Figure 1, we can always find a repeating unit, i.e. a unit cell, to describe the infinite periodic structure of a material. A unit cell that includes $N$ atoms can be fully described by 3 lists: 1) atom types $\boldsymbol{A} = (a_0, ..., a_N) \in \mathbb{A}^N$, where $\mathbb{A}$ denotes the set of all chemical elements; 2) atom coordinates $\boldsymbol{X} = (\boldsymbol{x}_0, ..., \boldsymbol{x}_N) \in \mathbb{R}^{N \times 3}$; and 3) periodic lattice $\boldsymbol{L} = (\boldsymbol{l}_1, \boldsymbol{l}_2, \boldsymbol{l}_3) \in \mathbb{R}^{3 \times 3}$. The periodic lattice defines the periodic translation symmetry of the material. Given $\boldsymbol{M} = (\boldsymbol{A}, \boldsymbol{X}, \boldsymbol{L})$, the infinite periodic structure can be represented as,

$$\{(a_i', \boldsymbol{x}_i') | a_i' = a_i, \boldsymbol{x}_i' = \boldsymbol{x}_i + k_1 \boldsymbol{l}_1 + k_2 \boldsymbol{l}_2 + k_3 \boldsymbol{l}_3, k_1, k_2, k_3 \in \mathbb{Z}\}, \tag{1}$$

where $k_1, k_2, k_3$ are any integers that translate the unit cell using $\boldsymbol{L}$ to tile the entire 3D space.

The chemical composition of a material denotes the ratio of different elements that the material is composed of. Given the atom types of a material with $N$ atoms $\boldsymbol{A} \in \mathbb{A}^N$, the composition can be represented as $\boldsymbol{c} \in \mathbb{R}^{|\mathbb{A}|}$, where $\boldsymbol{c}_i > 0$ denotes the percentage of atom type $i$ and $\sum_i \boldsymbol{c}_i = 1$. For example, the composition of diamond in Figure 1 has $\boldsymbol{c}_6 = 1$ and $\boldsymbol{c}_i = 0$ for $i \neq 6$ because 6 is the atomic number of carbon.

**Invariances for materials.** The structure of a material does not change under several invariances. 1) *Permutation invariance*. Exchanging the indices of any pair of atoms will not change the material. 2) *Translation invariance*. Translating the atom coordinates $\boldsymbol{X}$ by an arbitrary vector will not change the material. 3) *Rotation invariance*. Rotating $\boldsymbol{X}$ and $\boldsymbol{L}$ together by an arbitrary rotation matrix will not change the material. 4) *Periodic invariance*. There are infinite different ways of choosing unit cells with different shapes and sizes, e.g., obtaining a bigger unit cell as an integer multiplier of a smaller unit cell using integer translations. The material will again not change given different choices of unit cells.

**Multi-graph representation for materials.** Materials can be represented as a directed multi-graph $\mathcal{G} = \{\mathcal{V}, \mathcal{E}\}$ to encode the periodic structures following (Wells et al., 1977; O'Keeffe & Hyde, 1980; Xie & Grossman, 2018), where $\mathcal{V} = \{v_1, ..., v_N\}$ is the set of nodes representing atoms and $\mathcal{E} = \{e_{ij,(k_1,k_2,k_3)} | i, j \in \{1, ..., N\}, k_1, k_2, k_3 \in \mathbb{Z}\}$ is the set of edges representing bonds. $e_{ij,(k_1,k_2,k_3)}$ denotes a directed edge from node $i$ at the original unit cell to node $j$ at the cell translated by $k_1 \boldsymbol{l}_1 + k_2 \boldsymbol{l}_2 + k_3 \boldsymbol{l}_3$ (in Figure 1 right, $(k_1, k_2, k_3)$ are labeled on top of edges). For materials, there is no unique way to define edges (bonds) and the edges are often computed using k-nearest neighbor (KNN) approaches under periodicity or more advanced methods such as CrystalNN (Pan et al., 2021). Given this directed multi-graph, message-passing neural networks and SE(3)-equivariant networks can be used for the representation learning of materials.

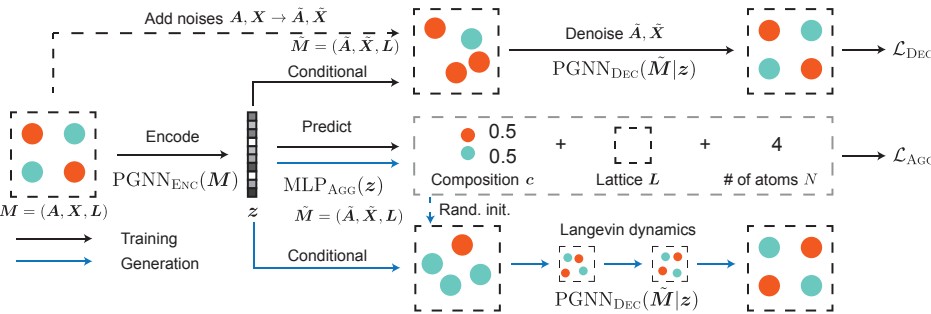

Figure 2: Overview of the proposed CDVAE approach.

## 3.2 PROBLEM DEFINITION AND ITS PHYSICAL ORIGIN

Our goal is to generate novel, stable materials $\boldsymbol{M} = (\boldsymbol{A}, \boldsymbol{X}, \boldsymbol{L}) \in \mathbb{A}^N \times \mathbb{R}^{N \times 3} \times \mathbb{R}^{3 \times 3}$. The space of stable materials is a subspace in $\mathbb{A}^N \times \mathbb{R}^{N \times 3} \times \mathbb{R}^{3 \times 3}$ that satisfies the following constraints. 1) The materials lie in the local minimum of the energy landscape defined by quantum mechanics, with respect to the atom coordinates and lattice, i.e. $\partial E / \partial \boldsymbol{X} = \boldsymbol{0}$ and $\partial E / \partial \boldsymbol{L} = \boldsymbol{0}$. 2) The material is globally stable and thus cannot decompose into nearby phases. Global stability is strongly related to bonding preferences between neighboring atoms. For example, in $SiO_2$, each Si is surrounded by 4 O and each O is surrounded by 2 Si. This configuration is caused by the stronger bonding preferences between Si-O than Si-Si and O-O.

Generally, finding novel, stable materials requires very expensive random search and quantum mechanical simulations. To bypass this challenge, we aim to learn a generative model $p(\boldsymbol{M})$ from the empirical distribution of experimentally observed stable materials. A successful generative model will be able to generate novel materials that satisfy the above constraints, which can then be verified using quantum mechanical simulations.

## 3.3 DIFFUSION MODELS

Diffusion models are a new class of generative models that have recently shown great success in generating high-quality images (Dhariwal & Nichol, 2021), point clouds (Cai et al., 2020; Luo & Hu, 2021), and molecular conformations (Shi et al., 2021). There are several different types of diffusion models including diffusion probabilistic models (Sohl-Dickstein et al., 2015), noise-conditioned score networks (NCSN) (Song & Ermon, 2019), and denoising diffusion probabilistic models (DDPM) (Ho et al., 2020). We follow ideas from the NCSN (Song & Ermon, 2019) and learn a score network $\boldsymbol{s_\theta}(\boldsymbol{x})$ to approximate the gradient of a probability density $\nabla_{\boldsymbol{x}} p(\boldsymbol{x})$ at different noise levels. Let $\{\sigma_i\}_{i=1}^L$ be a sequence of positive scalars that satisfies $\sigma_1 / \sigma_2 = ... = \sigma_{L-1} / \sigma_L > 1$. We define the data distribution perturbed by Gaussian noise $\sigma$ as $q_\sigma(\boldsymbol{x}) = \int p_{\text{data}}(\boldsymbol{t}) \mathcal{N}(\boldsymbol{x}|\boldsymbol{t}, \sigma^2 I) \, d\boldsymbol{t}$. The goal of NCSN is to learn a score network to jointly estimate the scores of all perturbed data distributions, i.e. $\forall \sigma \in \{\sigma_i\}_{i=1}^L : \boldsymbol{s_\theta}(\boldsymbol{x}, \sigma) \approx \nabla_{\boldsymbol{x}} q_\sigma(\boldsymbol{x})$. During generation, NCSN uses an annealed Langevin dynamics algorithm to produce samples following the gradient estimated by the score network with a gradually reduced noise level.

## 4 PROPOSED METHOD

Our approach generates new materials via a two-step process: 1) We sample a $\boldsymbol{z}$ from the latent space and use it to predict 3 aggregated properties of a material: composition ($\boldsymbol{c}$), lattice ($\boldsymbol{L}$), and number of atoms ($N$), which are then used to randomly initialize a material structure $\tilde{\boldsymbol{M}} = (\tilde{\boldsymbol{A}}, \tilde{\boldsymbol{X}}, \boldsymbol{L})$. 2) We perform Langevin dynamics to simultaneously denoise $\tilde{\boldsymbol{X}}$ and $\tilde{\boldsymbol{A}}$ conditioned on $\boldsymbol{z}$ to improve both the local and global stability of $\tilde{\boldsymbol{M}}$ and generate the final structure of the new material.

To train our model, we optimize 3 networks concurrently using stable materials $\boldsymbol{M} = (\boldsymbol{A}, \boldsymbol{X}, \boldsymbol{L})$ sampled from the data distribution. 1) A periodic GNN encoder $\text{PGNN}_{\text{ENC}}(\boldsymbol{M})$ that encodes $\boldsymbol{M}$ into a latent representation $\boldsymbol{z}$. 2) A property predictor $\text{MLP}_{\text{AGG}}(\boldsymbol{z})$ that predicts the $\boldsymbol{c}$, $\boldsymbol{L}$, and $N$ of $\boldsymbol{M}$ from $\boldsymbol{z}$. 3) A periodic GNN decoder $\text{PGNN}_{\text{DEC}}(\tilde{\boldsymbol{M}}|\boldsymbol{z})$ that denoises both $\tilde{\boldsymbol{X}}$ and $\tilde{\boldsymbol{A}}$ conditioned

on $z$. For 3), the noisy structure $\tilde{M} = (\tilde{A}, \tilde{X}, L)$ is obtained by adding different levels of noise to $X$ and $A$. The noise schedules are defined by the predicted aggregated properties, with the motivation of simplifying the task for our decoder from denoising an arbitrary random structure from over $\sim 100$ elements to a constrained random structure from predicted properties. We train all three networks together by minimizing a combined loss including the aggregated property loss $\mathcal{L}_{\text{AGG}}$, decoder denoising loss $\mathcal{L}_{\text{DEC}}$, and a KL divergence loss $\mathcal{L}_{\text{KL}}$ for the VAE.

To capture the interactions across periodic boundaries, we employ a multi-graph representation (section 3.1) for both $M$ and $\tilde{M}$. We also use SE(3) equivariant GNNs adapted with periodicity as both the encoder and the decoder to ensure the permutation, translation, rotation, and periodic invariances of our model. The CDVAE is summarized in Figure 2 and we explain the individual components of our method below. The implementation details can be found in Appendix B.

**Periodic material encoder.** $\text{PGNN}_{\text{ENC}}(M)$ encodes a material $M$ as a latent representation $z \in \mathbb{R}^D$ following the reparameterization trick in VAE (Kingma & Welling, 2014). We use the multi-graph representation (refer to section 3.1) to encode $M$, and $\text{PGNN}_{\text{ENC}}$ can be parameterized with an SE(3) invariant graph neural network.

**Prediction of aggregated properties.** $\text{MLP}_{\text{AGG}}(z)$ predicts 3 aggregated properties of the encoded material from its latent representation $z$. It is parameterized by 3 separate multilayer perceptrons (MLPs). 1) Composition $c \in \mathbb{R}^{|\mathbb{A}|}$ is predicted by minimizing the cross entropy between the ground truth composition and predicted composition, i.e. $-\sum_i p_i \log c_i$. 2) Lattice $L \in \mathbb{R}^{3 \times 3}$ is reduced to 6 unique, rotation invariant parameters with the Niggli algorithm (Grosse-Kunstleve et al., 2004), i.e., the lengths of the 3 lattice vectors, the angles between them, and the values are predicted with an MLP after being normalized to the same scale (Appendix B.1) with an $L_2$ loss. 3) Number of atoms $N \in \{1, 2, ...\}$ is predicted with a softmax classification loss from the set of possible number of atoms. $\mathcal{L}_{\text{AGG}}$ is a weighted sum of the above 3 losses.

**Conditional score matching decoder.** $\text{PGNN}_{\text{DEC}}(\tilde{M}|z)$ is a PGNN that inputs a noisy material $\tilde{M}$ with type noises $\sigma_A$, coordinate noises $\sigma_X$, as well as a latent $z$, and outputs 1) a score $s_X(\tilde{M}|z; \sigma_A, \sigma_X) \in \mathbb{R}^{N \times 3}$ to denoise the coordinate for each atom towards its ground truth value, and 2) a probability distribution of the true atom types $p_A(\tilde{M}|z; \sigma_A, \sigma_X) \in \mathbb{R}^{N \times |\mathbb{A}|}$. We use a SE(3) graph network to ensure the equivariance of $s_X$ with respect to the rotation of $\tilde{M}$. To obtain the noisy structures $\tilde{M}$, we sample $\sigma_A$ and $\sigma_X$ from two geometric sequences of the same length: $\{\sigma_{A,j}\}_{j=1}^L, \{\sigma_{X,j}\}_{j=1}^L$, and add the noises with the following methods. For type noises, we use the type distribution defined by the predicted composition $c$ to linearly perturb true type distribution $\tilde{A} \sim (\frac{1}{1+\sigma_A} p_A + \frac{\sigma_A}{1+\sigma_A} p_c)$, where $p_{A,ij} = 1$ if atom $i$ has the true atom type $j$ and $p_{A,ij} = 0$ for all other $j$s, and $p_c$ is the predicted composition. For coordinate noises, we add Gaussian noises to the true coordinates $\tilde{X} \sim \mathcal{N}(X, \sigma_X^2 I)$.

$\text{PGNN}_{\text{DEC}}$ is parameterized by a SE(3) equivariant PGNN that inputs a multi-graph representation (section 3.1) of the noisy material structure and the latent representation. The node embedding for node $i$ is obtained by the concatenation of the element embedding of $\tilde{a}_i$ and the latent representation $z$, followed by a MLP, $h_i^0 = \text{MLP}(e_a(\tilde{a}_i) \| z)$, where $\|$ denotes concatenation of two vectors and $e_a$ is a learned embedding for elements. After $K$ message-passing layers, $\text{PGNN}_{\text{DEC}}$ outputs a vector per node that is equivariant to the rotation of $\tilde{M}$. These vectors are used to predict the scores, and we follow Song & Ermon (2019); Shi et al. (2021) to parameterize the score network with noise scaling: $s_X(\tilde{M}|z; \sigma_A, \sigma_X) = s_X(\tilde{M}|z)/\sigma_X$. The node representations $h_i^K$ are used to predict the distribution of true atom types, and the type predictor is the same at all noise levels: $p_A(\tilde{M}|z; \sigma_A, \sigma_X) = p_A(\tilde{M}|z)$, $p_A(\tilde{M}|z)_i = \text{softmax}(\text{MLP}(h_i^K))$.

**Periodicity influences denoising target.** Due to periodicity, a specific atom $i$ may move out of the unit cell defined by $L$ when the noise is sufficiently large. This leads to two different ways to define the scores for node $i$. 1) Ignore periodicity and define the target score as $x_i - \tilde{x}_i$; or 2) Define the target score as the shortest possible displacement between $x_i$ and $\tilde{x}_i$ considering periodicity, i.e. $d_{\min}(x_i, \tilde{x}_i) = \min_{k_1, k_2, k_3}(x_i - \tilde{x}_i + k_1 l_1 + k_2 l_2 + k_3 l_3)$. We choose 2) because the scores are the same given two different $\tilde{X}$ that are periodically equivalent, which is mathematically grounded for periodic structures, and empirically results in much more stable training.

The training loss for the decoder $\mathcal{L}_{\mathrm{DEC}}$ can be written as,

$$\frac{1}{2L}\sum_{j=1}^{L}\left[\mathbb{E}_{q_{\mathrm{data}(M)}}\mathbb{E}_{q_{\sigma_{\boldsymbol{A},j},\sigma_{\boldsymbol{X},j}}(\tilde{\boldsymbol{M}}|\boldsymbol{M})}\left(\left\|\boldsymbol{s}_{\boldsymbol{X}}(\tilde{\boldsymbol{M}}|\boldsymbol{z})-\frac{\boldsymbol{d}_{\min}(\boldsymbol{X},\tilde{\boldsymbol{X}})}{\sigma_{\boldsymbol{X},j}}\right\|_2^2+\frac{\lambda_{\mathrm{a}}}{\sigma_{\boldsymbol{A},j}}\mathcal{L}_{\mathrm{a}}(\boldsymbol{p}_{\boldsymbol{A}}(\tilde{\boldsymbol{M}}|\boldsymbol{z}),\boldsymbol{p}_{\boldsymbol{A}})\right)\right],$$

(2)

where $\lambda_{\mathrm{a}}$ denotes a coefficient for balancing the coordinate and type losses, $\mathcal{L}_{\mathrm{a}}$ denotes the cross entropy loss over atom types, $\boldsymbol{p}_{\boldsymbol{A}}$ denotes the true atom type distribution. Note that to simplify the equation, we follow the loss coefficients in Song & Ermon (2019) for different $\sigma_{\boldsymbol{X},j}$ and $\sigma_{\boldsymbol{A},j}$ and factor them into Equation 2.

**Material generation with Langevin dynamics.** After training the model, we can generate the periodic structure of material given a latent representation $\boldsymbol{z}$. First, we use $\boldsymbol{z}$ to predict the aggregated properties: 1) composition $\boldsymbol{c}$, 2) lattice $\boldsymbol{L}$, and 3) the number of atoms $N$. Then, we randomly initialize an initial periodic structure $(\boldsymbol{A}_0, \boldsymbol{X}_0, \boldsymbol{L})$ with the aggregated properties and perform an annealed Langevin dynamics (Song & Ermon, 2019) using the decoder, simultaneously updating the atom types and coordinates. During the coordinate update, we map the coordinates back to the unit cell at each step if atoms move out of the cell. The algorithm is summarized in Algorithm 1.

---

**Algorithm 1** Material Generation via Annealed Langevin Dynamics

---

1: **Input:** latent representation $\boldsymbol{z}$, type and coordinate noise levels $\{\sigma_{\boldsymbol{A}}\}$, $\{\sigma_{\boldsymbol{X}}\}$, step size $\epsilon$, number of sampling steps $T$
2: Predict aggregated properties $\boldsymbol{c}, \boldsymbol{L}, N$ from $\boldsymbol{z}$.
3: Uniformly initialize $\boldsymbol{X}_0$ within the unit cell by $\boldsymbol{L}$.
4: Randomly initialize $\boldsymbol{A}_0$ with $\boldsymbol{c}$.
5: **for** $j \leftarrow 1$ to $L$ **do**
6:     $\alpha_j \leftarrow \epsilon \cdot \sigma_{\boldsymbol{X},j}^2/\sigma_{\boldsymbol{X},L}^2$
7:     **for** $t \leftarrow 1$ to $T$ **do**
8:         $\boldsymbol{s}_{\boldsymbol{X},t} \leftarrow \boldsymbol{s}_{\boldsymbol{X}}(\boldsymbol{A}_{t-1}, \boldsymbol{X}_{t-1}, \boldsymbol{L}|\boldsymbol{z}; \sigma_{\boldsymbol{A},j}, \sigma_{\boldsymbol{X},j})$
9:         $\boldsymbol{p}_{\boldsymbol{A},t} \leftarrow \boldsymbol{p}_{\boldsymbol{A}}(\boldsymbol{A}_{t-1}, \boldsymbol{X}_{t-1}, \boldsymbol{L}|\boldsymbol{z}; \sigma_{\boldsymbol{A},j}, \sigma_{\boldsymbol{X},j})$
10:         Draw $\boldsymbol{X}_t^\epsilon \sim \mathcal{N}(0, \boldsymbol{I})$
11:         $\boldsymbol{X}_t' \leftarrow \boldsymbol{X}_{t-1} + \alpha_j \boldsymbol{s}_{\boldsymbol{X},t} + \sqrt{2\alpha_i}\boldsymbol{X}_t^\epsilon$
12:         $\boldsymbol{X}_t \leftarrow \mathrm{back\_to\_cell}(\boldsymbol{X}_t', \boldsymbol{L})$
13:         $\boldsymbol{A}_t = \mathrm{argmax}\, \boldsymbol{p}_{\boldsymbol{A},t}$
14:     $\boldsymbol{X}_0 \leftarrow \boldsymbol{X}_T, \boldsymbol{A}_0 \leftarrow \boldsymbol{A}_T$

---

**Connection between the gradient field and a harmonic force field.** The gradient field $\boldsymbol{s}_{\boldsymbol{X}}(\tilde{\boldsymbol{M}}|\boldsymbol{z})$ is used to update atom coordinates in Langevin dynamics via the force term, $\alpha_j \boldsymbol{s}_{\boldsymbol{X},t}$. In Appendix A, we show that $\alpha_j \boldsymbol{s}_{\boldsymbol{X},t}$ is mathematically equivalent to[2] a harmonic force field $F(\tilde{\boldsymbol{X}}) = -k(\tilde{\boldsymbol{X}} - \boldsymbol{X})$ when the noises are small, where $\boldsymbol{X}$ is the equilibrium position of the atoms and $k$ is a force constant. Harmonic force field, i.e. spring-like force field, is a simple yet general physical model that approximates the forces on atoms when they are close to their equilibrium locations. This indicates that our learned gradient field utilizes the harmonic approximation to approximate QM forces without any explicit force data and generates stable materials with this physically motivated inductive bias.

## 5 EXPERIMENTS

We evaluate multiple aspects of material generation that are related to real-world material discovery process. Past studies in this field used very different tasks and metrics, making it difficult to compare different methods. Building upon past studies (Court et al., 2020; Ren et al., 2020), we create a set of standard tasks, datasets, and metrics to evaluate and compare models for material generation. Experiment details can be found in Appendix D.

**Tasks.** We focus on 3 tasks for material generation. 1) *Reconstruction* evaluates the ability of the model to reconstruct the original material from its latent representation $\boldsymbol{z}$. 2) *Generation* evaluates the validity, property statistics, and diversity of material structures generated by the model. 3) *Property optimization* evaluates the model's ability to generate materials that are optimized for a specific property.

**Datasets.** We curated 3 datasets representing different types of material distributions. 1) **Perov-5** (Castelli et al., 2012a;b) includes 18928 perovskite materials that share the same structure but differ in composition. There are 56 elements and all materials have 5 atoms in the unit cell. 2) **Carbon-24** (Pickard, 2020) includes 10153 materials that are all made up of carbon atoms but differ in structures. There is 1 element and the materials have 6 - 24 atoms in the unit cells. 3) **MP-20** (Jain et al., 2013) includes 45231 materials that differ in both structure and composition. There are

---

[2]In fact, this is also true for the original formulation of NCSN (Song & Ermon, 2019)

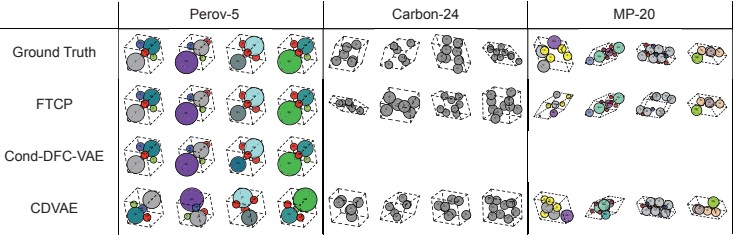

Figure 3: Reconstructed structures of randomly selected materials in the test set. Note our model reconstructs rotated (translated) version of the original material due to the SE(3) invariance.

Table 1: Reconstruction performance.

| Method | Match rate (%) ↑ | | | RMSE ↓ | | |
|---|---|---|---|---|---|---|
| | Perov-5 | Carbon-24 | MP-20 | Perov-5 | Carbon-24 | MP-20 |
| FTCP | **99.34** | **62.28** | **69.89** | 0.0259 | 0.2563 | 0.1593 |
| Cond-DFC-VAE | 51.65 | – | – | 0.0217 | – | – |
| CDVAE | 97.52 | 55.22 | 45.43 | **0.0156** | **0.1251** | **0.0356** |

89 elements and the materials have 1 - 20 atoms in the unit cells. We use a 60-20-20 random split for all of our experiments. Details regarding dataset curation can be found at Appendix C.

**Stability of materials in datasets.** Structures in all 3 datasets are obtained from QM simulations and all structures are at local energy minima. Most materials in Perov-5 and Carbon-24 are hypothetical, i.e. they may not have global stability (section 3.2) and likely cannot be synthesized. MP-20 is a realistic dataset that includes most experimentally known inorganic materials with at most 20 atoms in the unit cell, most of which are globally stable. A model achieving good performance in MP-20 has the potential to generate novel materials that can be experimentally synthesized.

**Baselines.** We compare CDVAE with the following 4 baselines, which include the latest coordinate-based, voxel-based, and 3D molecule generation methods. **FTCP** (Ren et al., 2020) is a crystal representation that concatenates real-space properties (atom positions, atom types, etc.) and Fourier-transformed momentum-space properties (diffraction pattern). A 1D CNN-VAE is trained over this representation for crystal generation. **Cond-DFC-VAE** (Court et al., 2020) encodes and generates crystals with 3D density maps, while employing several modifications over the previous Voxel-VAE (Hoffmann et al., 2019) method. However, the effectiveness is only demonstrated for cubic systems, limiting its usage to the Perov-5 dataset. **G-SchNet** (Gebauer et al., 2019) is an auto-regressive model that generates 3D molecules by performing atom-by-atom completion using SchNet (Schütt et al., 2018). Since G-SchNet is unaware of periodicity and cannot generate the lattice $L$. We adapt G-SchNet to our material generation tasks by constructing the smallest oriented bounding box with PCA such that the introduced periodicity does not cause structural invalidity. **P-G-SchNet** is our modified G-SchNet that incorporates periodicity. During training, the SchNet encoder inputs the partial periodic structure to predict next atoms. During generation, we first randomly sample a lattice $L$ from training data and autoregressively generate the periodic structure.

## 5.1 MATERIAL RECONSTRUCTION

**Setup.** The first task is to reconstruct the material from its latent representation. We evaluate reconstruction performance by matching the generated structure and the input structure for all materials in the test set. We use `StructureMatcher` from `pymatgen` (Ong et al., 2013), which finds the best match between two structures considering all invariances of materials. The match rate is the percentage of materials satisfying the criteria `stol=0.5`, `angle_tol=10`, `ltol=0.3`. The RMSE is averaged over all matched materials. Because the inter-atomic distances can vary significantly for different materials, the RMSE is normalized by $\sqrt[3]{V/N}$, roughly the average atom radius per material. Note G-SchNet is not a VAE so we do not evaluate its reconstruction performance.

**Results.** The reconstructed structures are shown in Figure 3 and the metrics are in Table 1. Since our model is SE(3) invariant, the generated structures may be a translated (or rotated) version of the ground truth structure. Our model has a lower RMSE than all other models, indicating its stronger capability to reconstruct the original stable structures. FTCP has a higher match rate than our model.

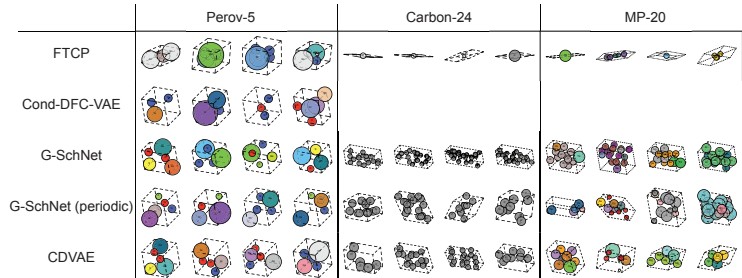

Figure 4: Structures sampled from $\mathcal{N}(0, 1)$ and filtered by the validity test.

Table 2: Generation performance[3].

| Method | Data | Validity (%) [4] ↑ | | COV (%) ↑ | | Property Statistics ↓ | | |
|---|---|---|---|---|---|---|---|---|
| | | Struc. | Comp. | R. | P. | $\rho$ | $E$ | # elem. |
| FTCP [5] | Perov-5 | 0.24 | 54.24 | 0.00 | 0.00 | 10.27 | 156.0 | 0.6297 |
| | Carbon-24 | 0.08 | – | 0.00 | 0.00 | 5.206 | 19.05 | – |
| | MP-20 | 1.55 | 48.37 | 4.72 | 0.09 | 23.71 | 160.9 | 0.7363 |
| Cond-DFC-VAE | Perov-5 | 73.60 | 82.95 | 73.92 | 10.13 | 2.268 | 4.111 | 0.8373 |
| G-SchNet | Perov-5 | 99.92 | **98.79** | 0.18 | 0.23 | 1.625 | 4.746 | **0.03684** |
| | Carbon-24 | 99.94 | – | 0.00 | 0.00 | 0.9427 | 1.320 | – |
| | MP-20 | 99.65 | 75.96 | 38.33 | **99.57** | 3.034 | 42.09 | 0.6411 |
| P-G-SchNet | Perov-5 | 79.63 | **99.13** | 0.37 | 0.25 | 0.2755 | 1.388 | 0.4552 |
| | Carbon-24 | 48.39 | – | 0.00 | 0.00 | 1.533 | 134.7 | – |
| | MP-20 | 77.51 | 76.40 | 41.93 | **99.74** | 4.04 | 2.448 | **0.6234** |
| CDVAE | Perov-5 | **100.0** | 98.59 | 99.45 | 98.46 | 0.1258 | 0.0264 | 0.0628 |
| | Carbon-24 | **100.0** | – | 99.80 | 83.08 | 0.1407 | 0.2850 | – |
| | MP-20 | **100.0** | 86.70 | 99.15 | 99.49 | 0.6875 | 0.2778 | 1.432 |

This can be explained by the fact that the same set of local structures can be assembled into different stable materials globally (e.g., two different crystal forms of ZnS). Our model is SE(3) invariant and only encodes local structures, while FTCP directly encodes the absolute coordinates and types of each atom. In Figure 5, we show that CDVAE can generate different plausible arrangements of atoms by sampling 3 Langevin dynamics with different random seeds from the same $z$. We note that this capability could be an advantage since it generates more diverse structures than simply reconstructing the original ones.

## 5.2 MATERIAL GENERATION

**Setup.** The second task is to generate novel, stable materials that are distributionally similar to the test materials. The only high-fidelity evaluation of stability of generated materials is to perform QM calculations, but it is computationally prohibitive to use QM for computing evaluation metrics. We developed several physically meaningful metrics to evaluate the validity, property statistics, and diversity of generated materials. 1) *Validity*. Following Court et al. (2020), a structure is valid as long as the shortest distance between any pair of atoms is larger than 0.5 Å, which is a relative weak criterion. The composition is valid if the overall charge is neutral as computed by SMACT (Davies et al., 2019). 2) *Coverage (COV)*. Inspired by Xu et al. (2021a); Ganea et al. (2021), we define two coverage metrics, COV-R (Recall) and COV-P (Precision), to measure the similarity between ensembles of generated materials and ground truth materials in test set. Intuitively, COV-R measures the percentage of ground truth materials being correctly predicted, and COV-P measures the percentage of predicted materials having high quality (details in Appendix G). 3) *Property statistics*. We compute the earth mover's distance (EMD) between the property distribution of generated materials and test materials. We use density ($\rho$, unit $g/cm^3$), energy predicted by an independent GNN ($E$, unit $eV/atom$), and number of unique elements (# elem.) as our properties. Validity and coverage are computed over 10,000 materials randomly sampled from $\mathcal{N}(0, 1)$. Property statistics is computed over 1,000 valid materials randomly sampled from those that pass the validity test.

---

[3]Some metrics unsuitable for specific datasets have "–" values in the table (explained in Appendix D.1).

[4]For comparison, the ground truth structure validity is 100.0 % for all datasets, and ground truth composition validity is 98.60 %, 100.0 %, 91.13 % for Perov-5, Carbon-24, and MP-20.

[5]Due to the low validity of FTCP, we instead randomly generate 100,000 materials from $\mathcal{N}(0, 1)$ and use 1,000 materials from those valid ones to compute diversity and property statistics metrics.

Table 3: Property optimization performance.

| Method | Perov-5 | | | Carbon-24 | | | MP-20 | | |
|---|---|---|---|---|---|---|---|---|---|
| | SR5 | SR10 | SR15 | SR5 | SR10 | SR15 | SR5 | SR10 | SR15 |
| FTCP | 0.06 | 0.11 | 0.16 | 0.0 | 0.0 | 0.0 | 0.02 | 0.04 | 0.05 |
| Cond-DFC-VAE | **0.55** | 0.64 | 0.69 | – | – | – | – | – | – |
| CDVAE | 0.52 | **0.65** | **0.79** | 0.0 | **0.06** | **0.06** | **0.78** | **0.86** | **0.90** |

**Results.** The generated structures are shown in Figure 4 and the metrics are in Table 2. Our model achieves a higher validity than FTCP, Cond-DFC-VAE, and P-G-SchNet, while G-SchNet achieves a similar validity as ours. The lower structural validity in P-G-SchNet than G-SchNet is likely due to the difficulty of avoiding atom collisions during the autoregressive generation inside a finite periodic box. On the contrary, our G-SchNet baseline constructs the lattice box after the 3D positions of all atoms are generated, and the construction explicitly avoids introducing invalidity. Furthermore, our model also achieves higher COV-R and COV-P than all other models, except in MP-20 our COV-P is similar to G-SchNet and P-G-SchNet. These results indicate that our model generates both diverse (COV-R) and high quality (COV-P) materials. More detailed results on the choice of thresholds for COV-R and COV-P, as well as additional metrics can be found in Appendix G. Finally, our model also significantly outperforms all other models in the property statistics of density and energy, further confirming the high quality of generated materials. We observe that our method tends to generate more elements in a material than ground truth, which explains the lower performance in the statistics of # of elems. than G-SchNet. We hypothesize this is due to the non-Gaussian statistical structure of ground truth materials (details in Appendix D.3), and using a more complex prior, e.g., a flow-model-transformed Gaussian (Yang et al., 2019), might resolve this issue.

## 5.3 PROPERTY OPTIMIZATION

**Setup.** The third task is to generate materials that optimize a specific property. Following Jin et al. (2018), we jointly train a property predictor $F$ parameterized by an MLP to predict properties of training materials from latent $z$. To optimize properties, we start with the latent representations of testing materials and apply gradient ascent in the latent space to improve the predicted property $F(\cdot)$. After applying 5000 gradient steps with step sizes of $1 \times 10^{-3}$, 10 materials are decoded from the latent trajectories every 500 steps. We use an independently trained property predictor to select the best one from the 10 decoded materials. Cond-DFC-VAE is a conditional VAE so we directly condition on the target property, sample 10 materials, and select the best one using the property predictor. For all methods, we generate 100 materials following the protocol above. We use the independent property predictor to predict the properties for evaluation. We report the success rate (SR) as the percentage of materials achieving 5, 10, and 15 percentiles of the target property distribution. Our task is to minimize formation energy per atom for all 3 datasets.

**Results.** The performance is shown in Table 3. We significantly outperform FTCP, while having a similar performance as Cond-DFC-VAE in Perov-5 (Cond-DFC-VAE cannot work for Carbon-24 and MP-20). Both G-SchNet and P-G-SchNet are incapable of property optimization [6]. We note that all models perform poorly on the Carbon-24 dataset, which might be explained by the complex and diverse 3D structures of carbon.

## 6 CONCLUSIONS AND OUTLOOK

We have introduced a Crystal Diffusion Variational Autoencoder (CDVAE) to generate the periodic structure of stable materials and demonstrated that it significantly outperforms past methods on the tasks of reconstruction, generation, and property optimization. We note that the last two tasks are far more important for material design than reconstruction because they can be directly used to generate new materials whose properties can then be verified by QM simulations and experiments. We believe CDVAE opens up exciting opportunities for the inverse design of materials for various important applications. Meanwhile, our model is just a first step towards the grand challenge of material design. We provide our datasets and evaluation metrics to the broader machine learning community to collectively develop better methods for the task of material generation.

---

[6]Very recently the authors published an improved version for conditional generation (Gebauer et al., 2021) but the code has not been released yet.

## REPRODUCIBILITY STATEMENT

We have made the following efforts to ensure reproducibility: 1) We provide our code at https://github.com/txie-93/cdvae; 2)We provide our data and corresponding train/validation/test splits at https://github.com/txie-93/cdvae/tree/main/data; 3) We provide details on experimental configurations in Appendix D.

## ACKNOWLEDGMENTS

We thank Peter Mikhael, Jason Yim, Rachel Wu, Bracha Laufer, Gabriele Corso, Felix Faltings, Bowen Jing, and the rest of the RB and TJ group members for their helpful comments and suggestions. The authors gratefully thank DARPA (HR00111920025), the consortium Machine Learning for Pharmaceutical Discovery and Synthesis (mlpds.mit.edu), and MIT-GIST collaboration for support.

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

## A  PROOF FOR THE CONNECTION TO A HARMONIC FORCE FIELD

We assume the loss in Equation 2 can be minimized to zero when the noises are small, meaning that

$$s_X(\tilde{A}, \tilde{X}, L|z) = \frac{d_{\min}(X, \tilde{X})}{\sigma_{X,j}}, \forall j > J, \tag{3}$$

where $\sigma_{X,j} \in \{\sigma_{X,j}\}_{j=1}^{L}$ and any noise smaller than $\sigma_{X,J}$ is considered as small.

The force term in the Langevin dynamics $\alpha_j s_{X,t}$ can then be written as

$$\alpha_j s_X(\tilde{A}, \tilde{X}, L|z; \sigma_{A,j}, \sigma_{X,j}) = \epsilon \cdot \sigma_{X,j}^2 / \sigma_{X,L}^2 \cdot s_X(\tilde{A}, \tilde{X}, L|z)/\sigma_{X,j} \tag{4}$$

$$= \epsilon \cdot \frac{\sigma_{X,j}^2}{\sigma_{X,L}^2} \cdot \frac{d_{\min}(X, \tilde{X})}{\sigma_{X,j}^2}, \forall j > J \tag{5}$$

$$= -\frac{\epsilon}{\sigma_{X,L}^2} d_{\min}(\tilde{X}, X), \forall j > J \tag{6}$$

If we write $\epsilon/\sigma_{X,L}^2 = k$, then,

$$\alpha_j s_X(\tilde{A}, \tilde{X}, L|z; \sigma_{A,j}, \sigma_{X,j}) = -k d_{\min}(\tilde{X}, X), \forall j > J \tag{7}$$

If the noises are small enough that atoms do not cross the periodic boundaries, then we have $d_{\min}(X, \tilde{X}) = X - \tilde{X}$. Therefore,

$$\alpha_j s_X(\tilde{A}, \tilde{X}, L|z; \sigma_{A,j}, \sigma_{X,j}) = -k(\tilde{X} - X), \forall j > J. \tag{8}$$

## B  IMPLEMENTATION DETAILS

### B.1  PREDICTION OF LATTICE PARAMETERS

There are infinitely many different ways of choosing the lattice for the same material. We compute the Niggli reduced lattice (Grosse-Kunstleve et al., 2004) with pymatgen (Ong et al., 2013), which is a unique lattice for any given material. Since the lattice matrix $L$ is not rotation invariant, we instead predict the 6 lattice parameters, i.e. the lengths of the 3 lattice vectors and the angles between them. We normalize the lengths of lattice vectors with $\sqrt[3]{N}$, where $N$ is the number of atoms, to ensure that the lengths for materials of different sizes are at the same scale.

### B.2  MULTI-GRAPH CONSTRUCTION

For the encoder, we use CrystalNN (Pan et al., 2021) to determine edges between atoms and build a multi-graph representation. For the decoder, since it inputs a noisy structure generated on the fly, the multi-graph must also be built on the fly for both training and generation, and CrystalNN is too slow for that purpose. We use a KNN algorithm that considers periodicity to build the decoder graph where $K = 20$ in all of our experiments.

### B.3  GNN ARCHITECTURE

We use DimeNet++ adapted for periodicity (Klicpera et al., 2020a;b) as the encoder, which is SE(3) invariant to the input structure. The decoder needs to output an vector per node that is SE(3) equivariant to the input structure. We use GemNet-dQ (Klicpera et al., 2021) as the decoder. We used implementations from the Open Catalysis Project (OCP) (Chanussot et al., 2021), but we reduced the size of hidden dimensions to 128 for faster training. The encoder has 2.2 million parameters and the decoder has 2.3 million parameters.

## C  DATASET CURATION

### C.1  PEROV-5

Perovskite is a class of materials that share a similar structure and have the general chemical formula $ABX_3$. The ideal perovskites have a cubic structure, where the site A atom sits at a corner position,

the site B atom sits at a body centered position and site X atoms sit at face centered positions. Perovskite materials are known for their wide applications. We curate the Perov-5 dataset from an open database that was originally developed for water splitting (Castelli et al., 2012a;b).

All 18928 materials in the original database are included. In the database, A, B can be any non-radioactive metal and X can be one or several elements from O, N, S, and F. Note that there can be multiple different X atoms in the same material. All materials in Perov-5 are relaxed using density functional theory (DFT), and their relaxed structure can deviate significantly from the ideal structures. A significant portion of the materials are not thermodynamically stable, i.e., they will decompose to nearby phases and cannot be synthesized.

## C.2 CARBON-24

Carbon-24 includes various carbon structures obtained via *ab initio* random structure searching (AIRSS) (Pickard & Needs, 2006; 2011) performed at 10 GPa.

The original dataset includes 101529 carbon structures, and we selected the 10% of the carbon structure with the lowest energy per atom to create Carbon-24. All 10153 structures in Carbon-24 are relaxed using DFT. The most stable structure is diamond at 10 GPa. All remaining structures are thermodynamically unstable but may be kinetically stable. Most of the structures cannot be synthesized.

## C.3 MP-20

MP-20 includes almost all experimentally stable materials from the Materials Project (Jain et al., 2013) with unit cells including at most 20 atoms. We only include materials that are originally from ICSD (Belsky et al., 2002) to ensure the experimental stability, and these materials represent the majority of experimentally known materials with at most 20 atoms in unit cells.

To ensure stability, we only select materials with energy above the hull smaller than 0.08 eV/atom and formation energy smaller than 2 eV/atom, following Ren et al. (2020). Differing from Ren et al. (2020), we do not constrain the number of unique elements per material. All materials in MP-20 are relaxed using DFT. Most materials are thermodynamcially stable and have been synthesized.

# D EXPERIMENT DETAILS

## D.1 REASONS FOR THE UNSUITABILITY OF SOME METRICS FOR SPECIFIC DATASETS

In Table 2, property statistics are computed by comparing the earth mover's distance between the property distribution of generated materials and ground truth materials. So, they are not meaningful for ground truth data.

Materials in Perov-5 have the same structure, so it is not meaningful to require higher structure diversity.

Materials in Carbon-24 have the same composition (carbon), so it is not meaningful to require higher composition diversity. In addition, all models have ∼100% composition validity, so it is not compared in the table.

## D.2 COMPOSITION VALIDITY CHECKER

We modified the charge neutrality checker from SMACT (Davies et al., 2019) because the original checker is not suitable for alloys. The checker is based on a list of possible charges for each element and it checks if the material can be charge neutral by enumerating all possible charge combinations. However, it does not consider that metal alloys can be mixed with almost any combination. As a result, for materials composed of all metal elements, we always assume the composition is valid in our validity checker.

For the ground truth materials in MP-20, the original checker gives a composition validity of ∼50%, which significantly underestimates the validity of MP-20 materials (because most of them are experimentally synthesizable and thus valid). Our checker gives a composition validity of ∼90%, which is far more reasonable. We note again that these checkers are all empirical and the only high-fidelity evaluation of material stability requires QM simulations.

### D.3 Non-Gaussian statistical structure of materials

The material datasets are usually biased towards certain material groups. For example, there are lots of lithium-containing materials in MP-20 because it started with battery research. We also find that our decoder tends to underfit the data distribution with a larger $\beta$ in Equation 9. We believe these observations indicate that the statistical structure of the ground truth materials are far from Gaussian. As a result, sampling from $\mathcal{N}(0, 1)$ may lead to out-of-distribution materials, which explains why our method tends to generate more elements per material than the ground truth.

### D.4 Hyperparameters and training details

The total loss can be written as,

$$\mathcal{L} = \mathcal{L}_{\text{AGG}} + \mathcal{L}_{\text{DEC}} + \mathcal{L}_{\text{KL}} = \lambda_{c}\mathcal{L}_{c} + \lambda_{L}\mathcal{L}_{L} + \lambda_{N}\mathcal{L}_{N} + \lambda_{X}\mathcal{L}_{X} + \lambda_{A}\mathcal{L}_{A} + \beta\mathcal{L}_{\text{KL}}. \quad (9)$$

We aim to keep each loss term at a similar scale. For all three datasets, we use $\lambda_{c} = 1, \lambda_{L} = 10, \lambda_{N} = 1, \lambda_{X} = 10, \mathcal{L}_{A} = 1$.

We tune $\beta$ between $0.01, 0.03, 0.1$ for all three datasets and select the model with best validation loss. For Perov-5, MP-20, we use $\beta = 0.01$, and for Carbon-24, we use $\beta = 0.03$.

For the noise levels in $\{\sigma_{A,j}\}_{j=1}^{L}, \{\sigma_{X,j}\}_{j=1}^{L}$, we follow Shi et al. (2021) and set $L = 50$. For all three datasets, we use $\sigma_{A,\max} = 5, \sigma_{A,\min} = 0.01, \sigma_{X,\max} = 10, \sigma_{X,\min} = 0.01$.

During the training, we use an initial learning rate of 0.001 and reduce the learning rate by a factor of 0.6 if the validation loss does not improve after 30 epochs. The minimum learning rate is 0.0001.

During the generation, we use $\epsilon = 0.0001$ and run Langevin dynamics for 100 steps at each noise level.

## E Visualization of multiple reconstructed structures

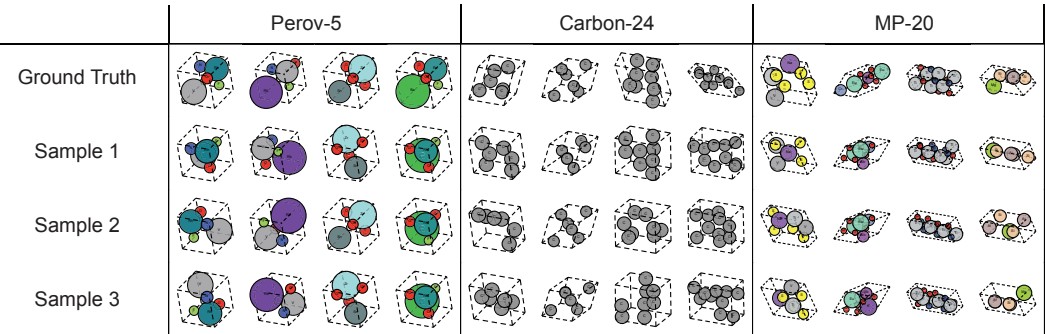

Figure 5: Different reconstructed structures from CDVAE from the same $z$, following 3 Langevin dynamics sampling with different random seeds.

## F Sampling speed for material generation

We summarize the speed for generating 10,000 materials for all models in Table 4. FTCP is significantly faster, but the quality of generated materials is very poor as shown in Table 2. Cond-DFC-VAE is faster than our method in Perov-5, but has a lower quality than our method and only works for cubic systems. It is also unclear how it will perform on larger materials in Carbon-24 and MP-20, because the compute increases cubically with the increased size of the density map. G-SchNet/P-G-SchNet have a comparable sampling time as our method, but have a lower quality. We also note that we did not optimize sampling speed in current work. It is possible to reduce sampling time by using fewer sampling steps without significantly influencing generation quality. There are also many recent works that aim to speed up the sampling process for diffusion models (Nichol & Dhariwal, 2021; Kong & Ping, 2021; Salimans & Ho, 2022).

Table 4: Time used for generating 10,000 materials on a single RTX 2080 Ti GPU.

|  | FTCP | Cond-DFC-VAE | G-SchNet | P-G-SchNet | CDVAE |
|---|---|---|---|---|---|
| Perov-5 | < 1 min | 0.5 h | 2.0 h | 2.0 h | 3.1 h |
| Carbon-24 | < 1 min | – | 6.2 h | 6.3 h | 5.3 h |
| MP-20 | < 1 min | – | 6.3 h | 6.3 h | 5.8 h |

## G  COVERAGE METRICS FOR MATERIAL GENERATION

Inspired by Xu et al. (2021a); Ganea et al. (2021), we define six metrics to compare two ensembles of materials: materials generated by a method $\{M_k\}_{k\in[1..K]}$, and ground truth materials in test data $\{M_l^*\}_{\in[1..L]}$.

We use the Euclidean distance of the CrystalNN fingerprint (Zimmermann & Jain, 2020) and normalized Magpie fingerprint (Ward et al., 2016) to define the structure distance and composition distance between generated and ground truth materials, respectively. They can be written as $D_{\text{struc.}}(M_k, M_l^*)$ and $D_{\text{comp.}}(M_k, M_l^*)$. We further define the thresholds for the structure and composition distance as $\delta_{\text{struc.}}$ and $\delta_{\text{comp.}}$, respectively.

Following the established classification metrics of Precision and Recall, we define the coverage metrics as:

$$\text{COV-R (Recall)} = \frac{1}{L}|\{l \in [1..L] : \exists k \in [1..K], D_{\text{struc.}}(M_k, M_l^*) < \delta_{\text{struc.}},$$
$$D_{\text{comp.}}(M_k, M_l^*) < \delta_{\text{comp.}}\}| \quad (10)$$

$$\text{AMSD-R (Recall)} = \frac{1}{L}\sum_{l\in[1..L]} \min_{k\in[1..K]} D_{\text{struc.}}(M_k, M_l^*) \quad (11)$$

$$\text{AMCD-R (Recall)} = \frac{1}{L}\sum_{l\in[1..L]} \min_{k\in[1..K]} D_{\text{comp.}}(M_k, M_l^*), \quad (12)$$

where COV is "Coverage", AMSD is "Average Minimum Structure Distance", AMCD is "Average Minimum Composition Distance", and COV-P (precision), AMSD-P (precision), AMCD-P (precision) are defined as in above equations, but with the generated and ground truth material sets swapped. The recall metrics measure how many ground truth materials are correctly predicted, while the precision metrics measure how many generated materials are of high quality (more discussions can be found in Ganea et al. (2021)).

We note several points on why we define the metrics in their current forms. 1) COV requires *both* structure and composition distances to be within the thresholds, because generating materials that are structurally close to one ground truth material and compositionally close to another is not meaningful. As a result, AMSD and AMCD are less useful than COV. 2) We use fingerprint distance, rather than RMSE from `StructureMatcher` (Ong et al., 2013), because the material space is too large for the models to generate enough materials to *exactly* match the ground truth materials. `StructureMatcher` first requires the compositions of two materials to exactly match, which will cause all models to have close-to-zero coverage.

For Perov-5 and Carbon-24, we choose $\delta_{\text{struc.}} = 0.2, \delta_{\text{comp.}} = 4$. For MP-20, we choose $\delta_{\text{struc.}} = 0.4, \delta_{\text{comp.}} = 10$. In Figure 6, Figure 7, Figure 8, we show how both COV-R and COV-P change by varying $\delta_{\text{struc.}}$ and $\delta_{\text{comp.}}$ in all three datasets.

Table 5: Full coverage metrics for the generation task.

| Method | Data | COV-R ↑ | AMSD-R ↓ | AMCD-R ↓ | COV-P ↑ | AMSD-P ↓ | AMCD-P ↓ |
|---|---|---|---|---|---|---|---|
| FTCP | Perov-5 | 0.00 | 0.7447 | 7.212 | 0.00 | 0.3582 | 3.390 |
| | Carbon-24 | 0.00 | 1.181 | 0.00 | 0.00 | 0.8822 | 24.16 |
| | MP-20 | 4.72 | 0.6542 | 9.271 | 0.09 | 0.1954 | 4.378 |
| Cond-DFC-VAE | Perov-5 | 73.92 | 0.1508 | 2.773 | 10.13 | 0.3162 | 4.257 |
| G-SchNet | Perov-5 | 0.18 | 0.5962 | 1.006 | 0.23 | 0.4259 | 1.3163 |
| | Carbon-24 | 0.00 | 0.5887 | 0.00 | 0.00 | 0.5970 | 0.00 |
| | MP-20 | 38.33 | 0.5365 | **3.233** | 99.57 | 0.2026 | 3.601 |
| P-G-SchNet | Perov-5 | 0.37 | 0.5510 | 1.0264 | 0.25 | 0.3967 | 1.316 |
| | Carbon-24 | 0.00 | 0.6308 | 0.00 | 0.00 | 0.8166 | 0.00 |
| | MP-20 | 41.93 | 0.5327 | 3.274 | **99.74** | 0.1985 | **3.567** |
| CDVAE | Perov-5 | **99.45** | **0.0482** | 0.6969 | **98.46** | **0.0593** | **1.272** |
| | Carbon-24 | **99.80** | **0.0489** | 0.00 | **83.08** | **0.1343** | 0.00 |
| | MP-20 | **99.15** | **0.1549** | 3.621 | **99.49** | **0.1883** | 4.014 |

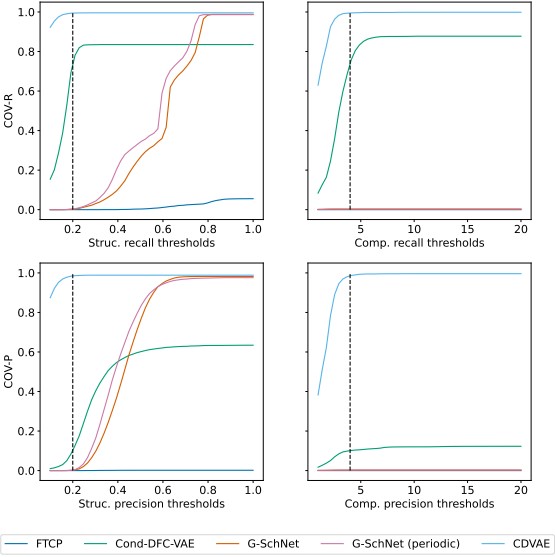

Figure 6: Change of COV-R and COV-P by varying $\delta_{\mathrm{struc.}}$ and $\delta_{\mathrm{comp.}}$ for Perov-5. Dashed line denotes the current chosen thresholds.

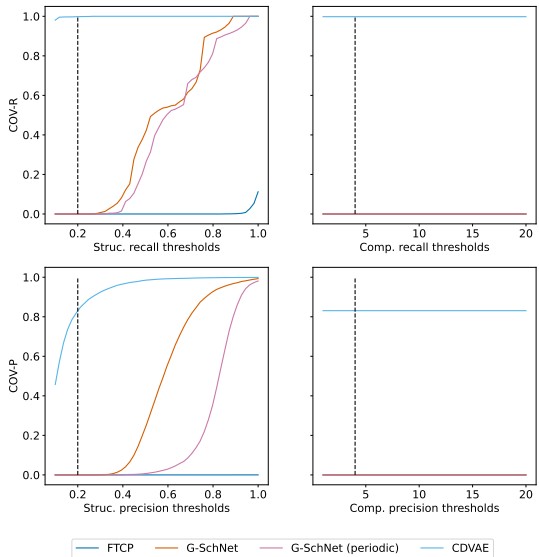

Figure 7: Change of COV-R and COV-P by varying $\delta_{\text{struc.}}$ and $\delta_{\text{comp.}}$ for Carbon-24. Dashed line denotes the current chosen thresholds.

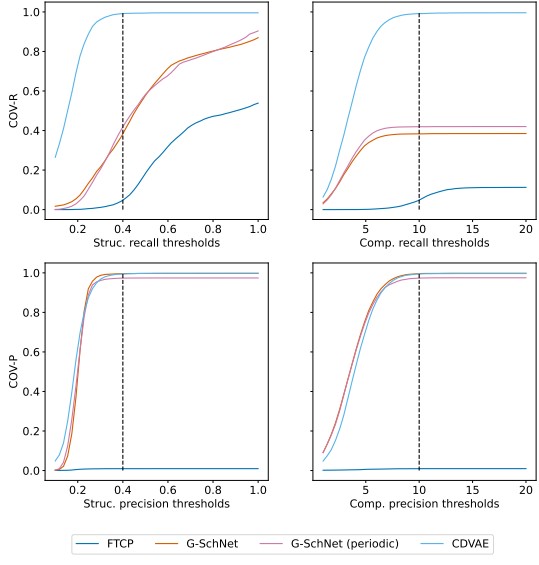

Figure 8: Change of COV-R and COV-P by varying $\delta_{\text{struc.}}$ and $\delta_{\text{comp.}}$ for MP-20. Dashed line denotes the current chosen thresholds.

