# OpenReview forum: "Crystal Diffusion Variational Autoencoder for Periodic Material Generation"
_ICLR.cc/2022/Conference — ICLR 2022 Poster_

### Official Review · Reviewer_5P5h · 2021-10-28

**Correctness:** 4
**Technical Novelty And Significance:** 3
**Empirical Novelty And Significance:** 4
**Recommendation:** 8
**Confidence:** 4

**Main Review:**

The approach is well motivated and novel. Being able to train a generative model without requiring large datasets of non-equilirbium configurations is very relevant considering the computational cost of electronic structure calculations.

The proposed CDVAE incorporates the necessary invariances and a connection to the harmonic approximation is shown. CDVAE shows state-of-the-art results in validity and diversity of the generated materials. The comparison G-SchNet is not ideal, as a proper adaption to materials would have included periodic boundary conditions. These should be relatively straightforward to implement since they are already available for SchNet. The evaluation could be improved by performing at least a limited number of QM calculations on a small random subset of the generated structures.

The optimization of the formation energy is another strong point of the paper.

**Summary Of The Paper:**

The paper proposes an approach to generate stable crystal structures using variational autoencoders without having to train on out-of-equilibrium data, which is scarce for materials. Beyond that, benchmark datasets and tasks for materials generation are proposed.

**Summary Of The Review:**

The proposed approach is a novel and relevant contribution to the domain of materials generation.

---

> ### Author Response · Authors · 2021-11-18
> **Reply to Reviewer 5P5h**
>
> We thank the reviewer for your helpful and constructive comments.
>
> **The comparison G-SchNet is not ideal, as a proper adaption to materials would have included periodic boundary conditions.**
>
> This is a great point. In our revised manuscript, we added a new baseline, P-G-SchNet, which incorporates periodicity into the G-SchNet framework. During training, the SchNet encoder inputs the partial periodic structure to predict next atoms.  During generation, we first randomly sample a lattice L from training data and autoregressively generate the periodic structure.
>
> Our results show that P-G-SchNet actually has a lower structure validity than the non-periodic G-SchNet, while having similar performance in other metrics. We think this is likely due to the difficulty of avoiding atom collisions during the autoregressive generation inside a finite periodic box. On the contrary, our G-SchNet baseline constructs the lattice box after the 3D positions of all atoms are generated, and the construction explicitly avoids introducing invalidity. We updated Table 2, Figure 4, the baseline section, and Sec. 5.2.
>
> **The evaluation could be improved by performing at least a limited number of QM calculations on a small random subset of the generated structures.**
>
> This is a great idea. We have started to work with our collaborators to perform some QM calculations. But creating a QM workflow and running the simulations require significant efforts and we have not obtained the results yet. We will certainly provide QM calculations in our future research.

---

> > ### Comment · Reviewer_5P5h · 2021-11-23
> > **Thank you for the response**
> >
> > I read the other reviews and find the author responses convincing. On this basis, I will keep my score unchanged.

---

### Official Review · Reviewer_Fg5a · 2021-10-30

**Correctness:** 3
**Technical Novelty And Significance:** 3
**Empirical Novelty And Significance:** 3
**Recommendation:** 5
**Confidence:** 4

**Main Review:**

Strengths:
- The tackled problem is specific but new and interesting for the machine learning research community, and the paper nicely provided 3 benchmarks to standardize the material generation problem.
- The paper is well-written and easy to follow. Both technical and empirical parts are clearly illustrated.
- The experiments are extensive to demonstrate the effectiveness across different datasets and metrics.

Weaknesses:
- Some detail of the "conditional score matching decoder" is confusing for me: how you sample the noisy atom type $\tilde{A} ~ (p_A +\sigma_A p_c)$. In my understanding, $p_A +\sigma_A p_c$ is not a valid categorical distribution, so I'm concerned with the sampling. Is there any softmax function imposed on it?
- Another concern is about the experiments. I appreciate the authors' efforts to provide several meaningful benchmarks, but the metrics in Sec 5.2 seem still very weak by just comparing validity, property, and fingerprints. There are some recent papers working on molecular conformation generation (samples are listed below), where the two conformation sets are compared by **coverage** and **matching** score. The main differences are the periodic and unconditional nature without graph input, but I think these two evaluations can still be easily adopted since you have also reported similar metrics in Sec 5.1. Could you provide more empirical justifications on the coverage and matching score?

 References for weakness 2:
- Xu, Minkai, et al. "Learning neural generative dynamics for molecular conformation generation." arXiv preprint arXiv:2102.10240 (2021).
- Ganea, Octavian-Eugen, et al. "GeoMol: Torsional Geometric Generation of Molecular 3D Conformer Ensembles." arXiv preprint arXiv:2106.07802 (2021).


**Summary Of The Paper:**

The paper proposed a new generative model for 3D periodic material molecular structure. They designed a special variational autoencoder, where the decoder is parameterized by the denoising score-matching framework. Experiments demonstrate the model can successfully generate valid and realistic material, and optimize desired properties.

**Summary Of The Review:**

The problem is specific but new to the general machine learning audience, and the proposed VAE model with equivariant score matching decoder is interesting for me. However, I still hold some technical confusion (weakness 1), and I think some important metrics from the highly related topic are missed in the material generation task, which is the most important empirical part of this paper. So currently I vote for a weak rejection.

---

> ### Author Response · Authors · 2021-11-18
> **Reply to Reviewer Fg5a**
>
> We thank the reviewer for helpful and constructive comments.
>
> **$p_A + \sigma_{A} p_c$ is not a valid categorical distribution. Is there any softmax function imposed on it?**
>
> Thanks for pointing this out. The actual distribution we sample from is $(\frac{1}{1 + \sigma_{A}} p_A + \frac{\sigma_{A}}{1 + \sigma_{A}} p_c)$. We neglected the normalization constant and have fixed the problem in the revised manuscript.
>
> **Could you provide more empirical justifications on the coverage and matching score?**
>
> This is a great suggestion. We have adopted the suggestion and now compute both COV-R (Recall) and COV-P (Precision) in our revised manuscript. In our definition of the coverage metrics, two materials match if both the structural and compositional fingerprint distances are within some thresholds. We use fingerprint distance, rather than the RMSE from StructureMatcher (Ong, et al., 2013), because the materials space is too large for the models to generate enough materials to exactly match the ground truth materials. If we use the StructureMatcher, the COV will be close to zero for all models. We also computed the matching scores, i.e. average minimum fingerprint distance, in Table 4 in Appendix E. The matching scores are in appendix because structural matching score or compositional matching score alone cannot fully describe material similarity. We have more detailed discussions in Appendix E.
>
> We updated Table 2 with the new COV metrics, updated Sec. 5.2, and added a new Appendix E with additional Table 4 and Figures 5-7. The new COV metrics are indeed stronger and the advantage of CDVAE is much clearer with the new metrics. Because COV-R already evaluates diversity and is probably better than our previous diversity metrics, we remove our previous diversity metrics in the revised manuscript. We note that G-SchNet has close-to-zero COV-R and COV-P in Perov-5 and Carbon-24, mainly due to the stricter thresholds we use to differentiate different methods in these two datasets. We investigated the influence of the choice of thresholds in Figs. 5-7, where the advantage of CDVAE is clear with different thresholds. We really appreciate the suggestions from the reviewer because it helped us to improve our work significantly.

---

> > ### Comment · Reviewer_Fg5a · 2021-11-27
> > **Thank you for answering questions and providing additional experiments.**
> >
> > The authors clarified the unclear parts and provided additional evaluations. I have also read other reviews (mainly from vK4U) and found some of them reasonable. Here is a follow-up question:
> > -  As indicated by vK4U, I also agree that this work is more like a workflow that combines multiple existing techniques. Regarding your special choice of using the denoising score-matching model (instead of flow-based or energy-based models), one main drawback should be the sampling speed. Typically, diffusion models suffer from an extremely long sampling time, which can be the bottleneck of the proposed model. Could you provide a comparison of the sampling time between your model and related baselines? I have a quick look again but didn't find it in the paper.

---

> > > ### Author Response · Authors · 2021-11-28
> > > **Thank you for your additional comments**
> > >
> > > We thank the reviewer for additional comments and suggestions.
> > >
> > > First, while flow models and EBMs are interesting and useful, it is nowhere trivial to devise them for inorganic periodic solids in a way that captures all constraints of this problem, and to our knowledge, there is no previous work on this. More concretely,
> > > - Flow models require an invertible function and a base distribution. It is unclear how to design them to satisfy the periodic requirements of 3D inorganic materials. In the latest work E-NFs from NeurIPS 2021 [1], the authors for the first time design a flow model to jointly generate the coordinate and atom types of 3D molecules. However, to satisfy translation invariance they explicitly assume $\sum_i x_i = 0$ and use a Gaussian distribution centered on the coordinate origin as base distribution. For infinite periodic systems, given that any lattice box may be arbitrarily translated in space without changing the underlying crystal structure, $\sum_i x_i = 0$ is no longer valid and the Gaussian distribution is also not suitable. This implies that E-NFs need a more fundamental change to properly model periodic materials, and we believe this requires a more extensive future work investigation.
> > > - For EBMs, it is still unclear how to generate atom types, coordinates, and lattice jointly. The most recent work (mentioned by Reviewer oZsg) designs an SE(3) equivariant EBM for protein folding [2]. However, it only generates the atom coordinates alone. Like in flow models, adapting EBMs for generating atom types and periodicity is also non-trivial and requires significant innovations.
> > > In comparison, diffusion score matching models do not suffer from these difficulties and they are conceptually easier to adapt to this challenging setting. Reversing the noising process is more natural and accessible than direct shot solutions. We leave extension of EBMs and flow models for future work.
> > >
> > > Second, for the generation speed, we agree that it is the major disadvantage of diffusion models. However, our method is still orders-of-magnitude faster than any non-ML approaches. Unlike in conformer generation, **there is no force field that is generally applicable for inorganic materials**. It means non-ML methods require extensive QM calculations to generate new energetically stable materials. For example, a recent work [3] generates ~20 stable materials after running 15031 QM calculations. Each QM calculation requires 1-5 hours. Therefore, our method can significantly accelerate material generation and assist existing expensive methods.
> > >
> > > Compared with baseline ML methods, we note that the **quality of generated materials from our model is significantly better than other methods** (see Tables 2,4 and Figures 5,6,7 from our paper), although the sampling time is longer in some cases. We summarize the generation time in the Table below and will include it in future revisions. FTCP is significantly faster, but the quality is very poor. G-SchNet/P-G-SchNet have a comparable sampling time as our method, but have a lower quality. Cond-DFC-VAE is faster than our method in Perov-5, but has a lower quality than our method and only works for Perov-5. It is also unclear how it will perform on larger materials in Carbon-24 and MP-20, because the compute increases cubicily with the increased size of the density map. We also note that **we did not optimize sampling speed in current work**. It is possible to reduce sampling time by using fewer sampling steps without significantly influencing generation quality. There are also many recent works that aim to speed up the sampling process, including [4, 5, 6].
> > >
> > > Table: time used for generating 10,000 materials on a single RTX 2080 Ti GPU.
> > >
> > > |  | FTCP | G-SchNet | P-G-SchNet | CDVAE | Cord-DFC-VAE |
> > > | -- | -- | -- | -- | -- | -- |
> > > | Perov-5 | < 1min | 2.0 h | 2.0 h | 3.1 h | 0.5 h |
> > > | Carbon-24 | < 1 min | 6.2 h | 6.3 h | 5.3 h | - |
> > > |MP-20| < 1min | 6.3 h | 6.3 h | 5.8 h | - |
> > >
> > >
> > > References:
> > >
> > > [1] Garcia Satorras, Victor, et al. "E (n) Equivariant Normalizing Flows." Advances in Neural Information Processing Systems 34 (2021).
> > >
> > > [2] Wu, Jiaxiang, et al. "SE (3)-Equivariant Energy-based Models for End-to-End Protein Folding." bioRxiv (2021).
> > >
> > > [3] Lu, Ziheng, et al. "Ab initio random structure searching for battery cathode materials." The Journal of Chemical Physics 154.17 (2021): 174111.
> > >
> > > [4] Nichol, Alex, and Prafulla Dhariwal. "Improved denoising diffusion probabilistic models." arXiv preprint arXiv:2102.09672 (2021).
> > >
> > > [5] Kong, Zhifeng, and Wei Ping. "On Fast Sampling of Diffusion Probabilistic Models." arXiv preprint arXiv:2106.00132 (2021).
> > >
> > > [6] “Progressive Distillation for Fast Sampling of Diffusion Models.” ICLR 2022 Conference, OpenReview.

---

### Official Review · Reviewer_oZsg · 2021-10-31

**Correctness:** 4
**Technical Novelty And Significance:** 3
**Empirical Novelty And Significance:** 3
**Recommendation:** 6
**Confidence:** 5

**Main Review:**

**strengths**
* The proposed score-based generative model for crystal structure generation, i.e., CDVAE, is novel. Although similar techniques have already been explored in other structure generation problems [1,2], I still believe the proposed model is quite novel as it addresses several important challenges in material structure generation such as **periodic boundary condition and composition prediction.**
* The proposed model is SE(3) equivariant and periodic boundary condition-sensitive, based on the recent advance on equivariant graph neural networks. It also considers the Periodicity influences on denoising target.
* The experimental design seems comprehensive, which may serve as standard benchmarks for inorganic crystal structure generation. The experimental results seems quite promising.
* The authors promise to provide the source code of the project after the review process.

**weaknesses**
* All the datasets included are inorganic. It would be very nice to see some initial results on organic crystal structure prediction.
* The lattice constant **L** is predicted from the latent vector and seems it is fixed during the decoding process.  You say that the structure of materials shoudl satisfy $\partial E / \partial L = 0$. How is this condition modeled in CDVAE? Can we update the lattice constant during decoding?
* The crystal structures often exhibit some symmetrcity, e.g. space group. I do not see any discussion related to this concept. Can CDVAE tackle the space group, or is this concept not applicable to the current datasets?

**Reference**
1. Learning gradient fields for molecular conformation generation
2. SE(3)-Equivariant Energy-based Models for End-to-End Protein Folding


**Summary Of The Paper:**

The paper proposes a score-based Diffusion Variational Autoencoder to generate the periodic structure of stable materials (crystal structure generation).

**Main contributions include**
* Creating a set of standard tasks, datasets and metrics to unify the evaluation process of inorganic crystal structure generation
* Proposing a **SE(3) equivariant, periodic boundary condition-sensitive** decoder to generate crystal structure structures based on a latent vector.



**Summary Of The Review:**

Overall, I think the submission is of high quality and reaches the bar of ICLR.
It would be nice if the authors could address the concerns listed above.

---

> ### Author Response · Authors · 2021-11-18
> **Reply to Reviewer oZsg**
>
> We thank the reviewer for helpful and constructive comments.
>
> **All the datasets included are inorganic. It would be very nice to see some initial results on organic crystal structure prediction.**
>
> This is an interesting point. Organic crystal structure generally refers to the prediction of crystal structure given the 2D molecular graph of organic molecules. This setting is different from ours because we do not condition on graphs or chemical composition. From our perspective, predicting organic crystal structure requires the model to learn two tasks together: 1) conformer structure prediction given the molecular 2D graph; 2) prediction of how different conformers will pack together. The second task is related to our problem, but combining them naturally requires a hierarchical model. We believe organic crystal structure is a very interesting question and we aim to tackle it in future research.
>
> **The lattice constant L is fixed. Can we update the lattice constant during decoding?**
>
> Currently, the $\partial E / \partial L = 0$ condition is modelled by directly predicting the lattice constant $L$ by the aggregated property predictor. $L$ is indeed fixed during the decoding. We do not update it during the decoding because $X$ and $L$ are strongly correlated due to periodicity. A slight change in $L$ will have a large influence on the pairwise distances/angles between all atoms, making the learning process unstable. We hope to improve this point in our future research.
>
> **Can CDVAE tackle the space group?**
>
> This is a great question. Currently, CDVAE do not use any space group information, i.e. it treats all crystals in the P1 space group. One potential way to incorporate space group information is to have another space group predictor and condition the generation on space groups. After the space group is predicted, we can constrain the generation of materials using the symmetry operations in each space group. We believe this is certainly an interesting direction for future research.
>
> **References**
>
> Thanks for mentioning these references. We’ve cited the references in our related work section.

---

> > ### Comment · Reviewer_oZsg · 2021-11-30
> > **Thank you for your reply.**
> >
> > The authors actually have pointed out several future directions, which I believe are quite valuable.
> > I have read other reviewers' comments, and for now, I would like to keep my original score.

---

### Official Review · Reviewer_vK4U · 2021-11-03

**Correctness:** 3
**Technical Novelty And Significance:** 2
**Empirical Novelty And Significance:** 3
**Recommendation:** 3
**Confidence:** 4

**Details Of Ethics Concerns:**

no ethics concerns

**Main Review:**

**Strength**
- This paper proposes an entire workflow to tackle crystal materials generation. It is an important step towards the grand challenge of materials design.
- The authors demonstrate the proposed method by using several datasets and baselines. Although the performance can be improved and some metrics are not very reasonable, these efforts can be identified as benchmarks for further exploration in the AI/ML community.

**Weakness**

General comments:
- The technical contribution on AI/ML side is probably marginal. Multiple recent models, like SE(3) with PGNNs, diffusion models (NCSN), CrystalNN, DimeNet++, GemNet-dQ, are simply used and combined without improvements. Although building the workflow is not trivial, I did not get an overall insight, e.g., why do you choose such models?

- Even though this paper is probably classified as an application paper, the contribution to the domain area may not be very substantial. First, the problem of crystal generation has been posted before and several studies already exist. Second, It is difficult to evaluate the success of the paper in crystal generation as it lacks good absolute metrics or a clear performance goal for a successful model. This approach does outperform existing methods in certain areas, but these existing methods are either too limited or too underperforming themselves to serve as a sufficiently challenging benchmark.

Detailed comments:

- The authors claim to generate stable structures due to some physical or chemical motivations. However, I did not get the insight that why stable data distribution will guarantee a stable structure generation via noising and denoising. Does it depend on how do you define the stable structure, it is statistical stable or physical stable?  I suppose that is the reason why you use the NCSN model for generation tasks but it might be easy to follow the intuition.  Why not using energy-based models [1] or flow-based models [2, 3], which are widely used for molecular generation?

[1] Liu, Meng, Keqiang Yan, Bora Oztekin, and Shuiwang Ji. "GraphEBM: Molecular graph generation with energy-based models." arXiv preprint arXiv:2102.00546 (2021).

[2] Shi, C., Xu, M., Zhu, Z., Zhang, W., Zhang, M. and Tang, J., 2020. Graphaf: a flow-based autoregressive model for molecular graph generation. arXiv preprint arXiv:2001.09382.

[3] Luo, Youzhi, Keqiang Yan, and Shuiwang Ji. "GraphDF: A discrete flow model for molecular graph generation." arXiv preprint arXiv:2102.01189 (2021).

- If NCSN works better than the other generative models, how do you define the sensitive noise parameters? The suggested noise levels are specifically designed for image generation, but do they work well for materials generation as well? If not, do you provide an ablation study to show the difference? Since now I do not see a code submission, I am curious about the reproducibility because the workflow looks pretty complex consisting of multiple-step training, rather than a straightforward end-to-end generation.

- Another weakness is the reconstruction loss from the VAE architecture. Visually, it is difficult to tell in Figure 3 if the reconstruction structures have shifted significantly or not. How big an impact does this reconstruction loss have on properties such as formation energy, for example? Is it also possible for elements to change upon reconstruction? In Figure 3 on the MP-20 dataset, it looks like the element has shifted for the second structure; is this really the case or a plotting error?

- A characteristic of this method is that the crystal parameters (composition and # of atoms, crystal lattice) are decoupled and predicted separately from the atomic structure. In addition, the generation is from random initiation, how do you define the upper and lower bound of your initialization, for different parameters? We know composition, # of atoms, lattice parameter share different scales and they are a mixture of continuous and discrete, do you normalize them? Have you considered dequantization?  It would be helpful to understand how to set up the dataset if these details can be provided.

- Validity criterion is likely too weak and likely not useful in evaluating generation performance, as most structures should not have interatomic distances anywhere near 0.5 Angstroms. Charge neutrality may also not be a good criterion due to the existence of defect structures or materials such as electrides.

- Authors use three existing datasets from the literature, containing (1) perovskites, (2) carbon, (3) and inorganic crystals. From what I can tell, the main contribution to these datasets by the authors is truncating the inorganic crystal dataset to a maximum of 20 atoms. Is 45000 for the inorganic crystal dataset a sufficiently large number of data for this type of problem, and to serve as a benchmark dataset? For the perovskite dataset, is it really necessary to use generative models here since they have a fixed ABO3 composition and do not have structural diversity, which defeats the purpose of a 3-D crystal generation dataset.

- Property optimization is not a fair comparison since the author’s method chooses the best out of ten, whereas other methods like Cond-DFC-VAE only use a single sample from conditioned generation. More importantly, property optimization based on optimized latent space should be able to generate realistic materials, but it might not be guaranteed due to several optimization issues. For example, gradient-based optimization in latent space (similar to JTVAE 2018 paper) may be trapped into local minima, do you consider a global optimizer, e.g., Bayesian optimization or CMA-ES?  I understand it also has scalability issues if the dimension of latent space is relatively high. In addition, some papers [GraphAF and conformation generation] use RL methods to generate a goal-oriented design.

- For property metrics, there is a high possibility of generating very out-of-distribution crystal samples from the training data; since the property is predicted by an independent GNN trained on the same data, there is a strong concern about the property predictions will be inaccurate and meaningless.

- It looks like G-SchNet is a strong contender in this paper, however the authors implement it without considering periodicity. This is not necessarily a failure of the G-SchNet method, but rather it is used improperly and cannot perform well, and allows the work in this paper to perform favorably compared to it.

- Scalability issue. The dataset used here is still a small-scale graph, does it work well for a large-scale crystal generation, such as more than 20 atoms in Materials Project?  Do you anticipate any limitations of the current workflow? If so, could you add more discussion about the existing limitations of CDVAE?



**Summary Of The Paper:**

This paper aims to address challenging stable crystal materials generation problems via diffusion variational autoencoder with graph representation learning. Several recent advances in generative models and GNN are combined together to develop the entire workflow from data distribution learning, prediction to sample generation and property optimization. The authors demonstrate their method by using three datasets and compare with three baseline methods.

**Summary Of The Review:**

The paper proposes to tackle challenging problems and provide a reasonably workable framework with several benchmarks. However, the core contributions from either the technical side or domain science side are not substantial and significant. Multiple pieces of work need more clear justification and explanation, and the metrics need to be improved with a more physically reasonable definition. I also expect to see a better comparison with the current baselines. Probably, the work is still ongoing so I recommend to reject it in the current form but expect to see more improvements in the near future.

---

> ### Author Response · Authors · 2021-11-18
> **Reply to Reviewer vK4U (part 1)**
>
> We thank the reviewer for detailed comments. We respectfully disagree with some of the raised concerns. We first address several main concerns, and then provide point-to-point responses.
>
> **Technical contribution to ML.** Our CDVAE architecture has multiple novel technical innovations that are necessary for material generation. 1) To solve the periodicity problem, we designed a custom denoising target for the NCSN that significantly improved training stability. We also incorporated periodicity into the Langevin dynamics generation process. These are non-trivial improvements to the original NCSN formulation. 2) Our NCSN decoder not only denoises coordinates but also denoises atom types jointly. This hybrid denoising differs from both the original NCSN and ConfGF (Shi et al., ICML 2021). 3) The forward diffusion process in our NCSN decoder is conditioned on the output of the aggregated property predictor, rather than fixed. This design significantly simplifies the task for our NCSN decoder and has not been reported before to the best of our knowledge.
>
> **Overall insight for our design.** Our architecture is strongly motivated by physics. The structure of stable materials are at the energy minimum defined by quantum mechanics (QM), so adding noise to the structure will increase its energy and produce a gradient in the opposite direction. The coordinate and type denoising processes in our NCSN decoder are motivated by local and global stability of materials, respectively (Sec. 3.2). In addition, we show a connection of NCSN to the harmonic approximation in physics at the end of Sec. 4.
>
> **Contribution to the domain.** Material generation and inverse design (i.e. the property optimization task) are long-standing challenges for materials discovery. There are review papers specifically discussing the significance of the problem [Noh, et al. Chemical Science 11.19 (2020): 4871-4881], and general review papers mentioning inverse design as an important problem [Butler, et al. Nature 559.7715 (2018): 547-555]. The problem of crystal generation has indeed been posted before, however without explicitly integrating important first-principles into these generative models, such as stability constraints, periodicity, Euclidean equivariances. Our model is, to our knowledge, the first to incorporate these geometric and physical constraints explicitly, showcasing empirically significant improvements over state-of-the-art methods. Reviewer 5P5h also mentions the significance of generating materials without using large datasets of non-equilibrium configurations. In addition, we design a set of datasets and evaluation metrics that are much more comprehensive than past studies.
>
> **Stronger evaluation metrics.** We agree with the reviewer that the previous metrics may have some weaknesses. We added two new, stronger evaluation metrics -- COV-R (Recall) and COV-P (Precision). These stronger metrics help to strengthen the significant improvements of our CDVAE over previous methods. For example, our model has a significantly higher COV-R and COV-P than all other methods including G-SchNet, except for the COV-P in MP-20 (updated Table 2). We note that G-SchNet has close-to-zero COV-R and COV-P in Perov-5 and Carbon-24, mainly due to the stricter thresholds we use to differentiate different methods in these two datasets. We investigated the influence of the choice of thresholds in Figs. 5-7, where the advantage of CDVAE is clear with different thresholds. These results indicate that CDVAE generates both diverse (COV-R) and high quality (COV-P) materials.
>
> **G-SchNet with periodicity as an additional baseline.** We have developed a version of G-SchNet that incorporates periodicity, we call P-G-SchNet, in its autoregressive generation. We found that P-G-SchNet actually has a lower structure validity than G-SchNet, with similar performance as G-SchNet in other metrics. We believe it is due to the difficulty of avoiding atom collisions during the autoregressive generation inside a finite periodic box. On the contrary, our G-SchNet baseline constructs the lattice box after the 3D positions of all atoms are generated, and the construction explicitly avoids introducing invalidity. The added baseline helps to further show the advantage of CDVAE over autoregressive models.

---

> > ### Author Response · Authors · 2021-11-18
> > **Reply to Reviewer vK4U (part 2)**
> >
> > Point-to-point responses:
> >
> > **Why does stable data distribution guarantee a stable structure? Is it statistically stable or physically stable?**
> >
> > In the stable data distribution, all materials must be at their equilibrium configurations, i.e. energy minimum defined by QM. Adding noises to the structure will increase its energy and produce a gradient (force) in the opposite direction. This force is approximated with noise conditioned scores in our model. Shi et al., ICML 2021 call this “pseudo-forces” in their ConfGF paper. We also show its connection to an Harmonic Force Field in physics in our paper, which may explain why it is a good physical inductive bias. Therefore, we believe our model learns not only statistical stability but also physical stability.
> >
> > **Why not using energy-based models [1] or flow-based models [2, 3]?**
> >
> > In addition to the physical motivation discussed above and in paper introduction and Sec. 3.2, we use NCSN for several reasons. 1) Unlike molecules, the structure of materials is periodic and has more long-range interdependencies. This makes it harder to use autoregressive models to generate materials which require a specific generation order. NCSN is non-autoregressive and can better handle these interdependencies. 2) We find that multiple levels of noise in NCSN are necessary to denoise an initial random structure. 3) It is easier to incorporate atom types denoising and periodicity in NCSN. 4) ConfGF was state-of-the-art in molecular conformer generation. We certainly believe it is possible to have EBMs and flow models for material generation as well. We’ve cited the mentioned papers in our related work section.
> >
> > **How do you define the sensitive noise parameters?**
> >
> > For the noise levels in our NCSN decoder, we follow ConfGF (Shi et al., ICML 2021) and set the number of noise levels to be 50. For all three datasets, we use the same set of noise parameters for all three datasets. The parameters are set by considering the size of the periodic lattice, and we didn’t tune the parameters extensively. We realize that we forgot to include these parameters in our previous version. We have added a new paragraph in our hyperparameter section in Appendix D.4 to include these details.
> >
> > **I do not see a code submission. I am curious about the reproducibility because the workflow looks pretty complex consisting of multiple-step training.**
> >
> > We’ve attached our code/data in a link that is only visible to Reviewers and Area Chairs, and we will publish it to the community after the paper decision is released. We note that the training is end-to-end because all 3 networks are trained together. The two-step generation, i.e. 1) generate random initial structure, 2) perform Langevin dynamics, is typical for diffusion models.
> >
> > **Concerns about Figure 3, the impact of reconstruction loss, whether atoms can change, and whether there is a plotting error.**
> >
> > Figure 3 is just for reference and the quantitative results are in Table 1. We didn’t investigate how different variants of reconstruction loss will influence the performance, and we plan to leave it for future research. But we note that the multiple levels of noises in our reconstruction loss is necessary for the model to work. The elements can indeed change upon reconstruction, and the atom type denoising is important for reconstruction. The element change in Figure 3 is not a plotting error. If a model fails to predict the correct atom type, it will lead to a low match rate. The matching criteria in Table 1 requires both atom types, coordinates, and lattice to match.
> >
> > **How to define the upper and lower bound of random initialization?**
> >
> > The initialization is purely determined by the aggregated property predictor. There are no additional parameters. The atom types are initialized using the predicted composition. The coordinates are initialized uniformly in the predicted lattice box.
> >
> > **Composition, # of atoms, lattice parameter are a mixture of continuous and discrete, do you normalize them? Have you considered dequantization? Details are helpful for setting up the dataset.**
> >
> > Both composition and number of atoms are predicted using cross entropy losses. The lattice parameters are continuous and we normalize them before training. We haven’t tried dequantization. But this is interesting, and we propose to leave it for further explorations. For the dataset, we’ve provided them in the code source link that is only visible to Reviewers and Area Chairs.

---

> > > ### Author Response · Authors · 2021-11-18
> > > **Reply to Reviewer vK4U (part 3)**
> > >
> > > **Validity criterion is likely too weak.**
> > >
> > > We agree that the validity criteria are not perfect, but evaluating the validity of materials without QM simulations is a long-standing challenge for materials design. The current metrics were proposed in the latest research [Court, et al. Journal of chemical information and modeling 60.10 (2020): 4518-4535] and [Davies, et al. Journal of Open Source Software 4.38 (2019): 1361]. Validity criteria are difficult mainly due to the significantly more diverse structures / elements in materials compared with small molecules. If the reviewer has any suggestions for a better validity criterion, we are more than happy to try. We want to add that COV-P and property statistics can provide additional information about the quality of generated materials. We also note that defect structures or electrides do not exist in our datasets.
> > >
> > > **What is the contribution in dataset curation? Is 45000 for the inorganic crystal dataset sufficiently large to serve as a benchmark dataset?**
> > >
> > > For all datasets, we perform data-cleaning to make sure all materials are stable (refer to “Stability of materials in datasets” in Sec. 5 and Appendix C). We used a few criteria including whether materials exist in ICSD, the energy above the hull, and formation energy to filter out unstable materials. In fact, the 45,000 materials are nearly all experimentally known materials with no more than 20 atoms in the unit cell. The number of known materials is much smaller than molecules.
> > >
> > > **For the perovskite dataset, is it really necessary to use generative models here since they have a fixed ABO3 composition and do not have structural diversity, which defeats the purpose of a 3-D crystal generation dataset.**
> > >
> > > The perovskite dataset is indeed easier than others. But the ABX3 perovskite structure has 5 distinct sites and can also be distorted. The model still needs to predict: 1) the correct site for each atom; 2) the distortion. Although Perov-5 is simple, only our model is able to achieve COV-R and COV-P > 90%. In addition, the perovskite dataset has real-world significance in materials design [Tao, et al. npj Computational Materials 7.1 (2021): 1-18.]
> > >
> > > **Property optimization is not a fair comparison for Cond-DFC-VAE. Gradient-based optimization in latent space (similar to JTVAE 2018 paper) may be trapped into local minima. Do you consider a global optimizer, e.g., Bayesian optimization or CMA-ES or RL?**
> > >
> > > This is a great point. We modified our protocol for Cond-DFC-VAE. As explained in revised Sec. 5.3, for Cond-DFC-VAE, we now sample 10 materials conditioned on the target property and select the best one using the property predictor. The process is repeated 100 times to generate the 100 materials for evaluation. This new protocol indeed increased their success rates to values similar to our model (updated Table 3). However, Cond-DFC-VAE cannot work for noncubic systems. The authors wrote “we are unable to recover the unit-cell angles in the noncubic case due to the differing radius of each site in the electron-density map.” “Further work will look at modifying the electron-density map encoding to better handle noncubic crystals.” Therefore, our model still has advantages in non-cubic systems. Both BO and RL are great ideas and will likely improve the performance. Due to the scope of this paper, we plan to leave them for future research and we think these are very exciting directions.
> > >
> > > **For property metrics, there is a high possibility of generating very out-of-distribution crystal samples and the independent GNN predictions may be inaccurate.**
> > >
> > > Since the GNN is independently trained with a different task from our encoder GNN, the out-of-distribution behavior will likely be different. In addition, we only use the independent GNN for the final evaluation, rather than optimizing our generator over it. So it is unlikely that our generator will exploit the independent GNN and generate very out-of-distribution materials. But we do acknowledge that this is a potential concern, which is also a well-known issue in the evaluation of molecule generative models. We are currently working with our collaborators to perform unbiased QM evaluations for property optimization. However, these evaluations are very expensive and require significant efforts to set up. We haven’t gotten the results yet.

---

> > > > ### Author Response · Authors · 2021-11-18
> > > > **Reply to Reviewer vK4U (part 4)**
> > > >
> > > > **It looks like G-SchNet is a strong contender in this paper, however the authors implement it without considering periodicity.**
> > > >
> > > > This is a valid concern. In our revised manuscript, we added a new baseline, P-G-SchNet, which incorporates periodicity into the G-SchNet framework. In revised Table 2, our results show that P-G-SchNet actually has a lower structure validity than the non-periodic G-SchNet. We think this is likely due to the difficulty of avoiding atom collisions during the autoregressive generation inside a finite periodic box. On the contrary, our G-SchNet baseline constructs the lattice box after the 3D positions of all atoms are generated, and the construction explicitly avoids introducing invalidity. In addition, our new, stronger metrics COV-R and COV-P further show that CDVAE significantly outperforms both G-SchNet and P-G-SchNet.
> > > >
> > > >
> > > > **Scalability issue. Does the model work for crystals with more than 20 atoms? Do you anticipate any limitations?**
> > > >
> > > > CDVAE uses PGNNs as encoder and decoder, so the learned model can generalize to larger crystals. Consequently, we believe CDVAE has an advantage in larger crystals than the baseline models, which lack the permutation invariances. We limit the MP-20 to no more than 20 atoms mainly to follow the FTCP paper [Ren, et al. arXiv preprint arXiv:2005.07609 (2020).] The baseline papers also said that their model cannot perform well in larger crystals. In addition, we note that MP-20 already covers 56.5% of all stable materials in the Materials Project. It is also generally easier to synthesize simpler crystals like those in MP-20 than more complex crystals.
> > > >
> > > > The limitations of CDVAE includes: 1) CDVAE currently generates more elements on average than G-SchNet. 2) CDVAE currently does not update the lattice during the Langevin dynamics. We are actively working on solving these limitations.

---

### Author Response · Authors · 2021-11-18
**General comments to all reviewers**

We thank all reviewers for their helpful and constructive comments. We made the following major revisions to the manuscript:

- To address the concerns that previous metrics may not be strong enough, we added two new stronger “coverage” metrics for generation: COV-R (Recall) and COV-P (Precision), which are widely used for molecule conformer generation. On these new metrics, our CDVAE model significantly outperforms all baselines. See updated Table 2, Sec. 5.2, Appendix E, Table 4, Figures 5-7.
- To address the concerns about G-SchNet not incorporating periodicity, we add a new baseline: P-G-SchNet, a modified G-SchNet that incorporates periodicity. We found that P-G-SchNet actually has a lower structure validity than G-SchNet, while performing similarly as G-SchNet in other metrics. The low structure validity is probably due to the difficulty of avoiding atom collisions during the autoregressive generation inside a finite periodic box. On the contrary, our G-SchNet baseline constructs the lattice box after the 3D positions of all atoms are generated, and the construction explicitly avoids introducing invalidity. See updated Table 2, Figure 4, the baseline section, and Sec. 5.2.

We also revised various other parts in our manuscript and appendix to address other comments by the reviewers. All changes are highlighted in red.

---

### Author Response · Authors · 2021-11-29
**General comments to all reviewers before the end of discussion period**

We thank the reviewers for all comments. We notice that there are some common concerns from Reviewer vK4U and Fg5a about the technical novelty of our method. We would like to take the opportunity to highlight the technical and empirical contributions of our paper again.

In the individual replies, we have added a new, stronger baseline, added multiple stronger evaluation metrics, significantly updated various parts of our manuscript (Tables 2, 3, Figures 4, 5, 6, Sec. 2, 4, 5, 5.2, 5.3, Appendix C3, D4, E). We greatly appreciate the reviewers’ comments, which help us improve our manuscript.

We have made a significant effort over the past 2 weeks to perform many additional experiments and answer all questions. We therefore humbly request that the Reviewer vK4U and Fg5a consider **engaging in a dialogue with us and share if there are any concerns left**.

Technical contributions:
- Our CDVAE architecture combines VAE, an aggregated property predictor, and NCSN to generate periodic materials.
- Our NCSN jointly denoises both atom types and coordinates, extending past methods like ConfGF which only denoises coordinates (Shi et al., ICML 2021)
- Multiple technical innovations to handle periodicity, including customized denoising targets that significantly improve training stability, dynamical multi-graph for periodic materials, handling periodicity in Langevin dynamics, etc. These have not been reported before to the extent of our knowledge.
- Forward diffusion process is conditioned on the aggregated property predictor, rather than fixed, which significantly simplifies the learning of reverse diffusion. This design is also not reported before to the extent of our knowledge.

Empirical contributions:
- We **significantly outperformed** 4 past state-of-the-art methods: 1) coordinate-based FTCP; 2) voxel-based Cond-DFC-VAE; 3) 3D molecule generation G-SchNet; 4) periodically adapted P-G-SchNet.
- We introduced a comprehensive set of new tasks and evaluation metrics for the challenging problem of material generation.
- The material generation problem is a long-standing challenge in materials design with real-world significance. There are review papers specifically for this problem [Noh, et al. Chemical Science 11.19 (2020): 4871-4881], and general review papers mentioning inverse design as an important problem [Butler, et al. Nature 559.7715 (2018): 547-555].
- We note that the problem of material generation is very challenging and it requires significant efforts to extend methods like EBMs or flow models to generate periodic materials (detailed discussion in our reply to Reviewer Fg5a). We have tried our best to include the latest, strongest approaches as our baselines.

---

### Decision · Program_Chairs · 2022-01-20

**Decision:**

Accept (Poster)

**Comment:**

This paper introduces a model, named Crystal Diffusion Variational Autoencoder (CDVAE), that can learn to sample valid material structures. It accounts for known symmetries (SE(3), permutation) of the structure via SE(3)-equivariant GNNs.

The proposed model is a complicated combination of many existing models / modeling techniques (VAEs, NCSNs, diffusion models) but it is not entirely ad hoc; the revised paper does a reasonable job in justifying the many different modeling choices made.
Existing model components, often designed in the context of molecule generation, do not account for the periodicity of the crystal's lattice structure; so this paper introduces modifications to account for this periodicity.

The paper evaluates the model on several datasets and also introduces new benchmarks that can be used for further research. The experimental results look promising but there are a few remaining clarity issues with the metrics used (cf. reviews).